

# Influences of sources and weather dynamics on atmospheric deposition of Se species and other trace elements

Esther S. Breuninger[1,2], Julie Tolu[1,2], Iris Thurnherr[3], Franziska Aemisegger[3], Aryeh Feinberg[4], Sylvain Bouchet[1,2], Jeroen E. Sonke[5], Véronique Pont[6], Heini Wernli[3] and Lenny H.E. Winkel[1,2]

[1]Eawag, Swiss Federal Institute of Aquatic Science and Technology, Ueberlandstrasse 133, 8600 Dübendorf
[2]Institute of Biogeochemistry & Pollutant Dynamics, ETH Zurich, 8092 Zurich, Switzerland
[3]Institute for Atmospheric and Climate Science, ETH Zurich, 8092 Zurich, Switzerland
[4]Institute for Data, Systems, and Society, Massachusetts Institute of Technology, Cambridge, MA 02142, USA
[5]Géosciences Environnement Toulouse, CNRS/IRD/Université de Toulouse, 31400 Toulouse, France
[6]LAERO, Université de Toulouse, CNRS, IRD, UT3, 31400 Toulouse, France

*Correspondence to*: Esther S. Breuninger, Lenny H.E. Winkel





**Abstract.**

Atmospheric deposition is an important source of the essential trace element selenium (Se) to terrestrial ecosystems and food chains. The fate of Se supplied to surface environments by atmospheric deposition strongly depends on total Se concentrations as well as its chemical form (speciation). However, the factors determining total Se and its speciation in atmospheric deposition remain poorly understood. Here, we applied different chemical measurements to aerosol samples taken at a weekly resolution over 5 years (2015-2019), as well as precipitation and cloud water samples taken during a field campaign

of two months in 2019 at Pic du Midi Observatory (French Pyrenees; 2877 m a.s.l.) and combined these observations with sophisticated modelling approaches. The high-altitude site enables the investigation of local and long-range elemental transport from both marine and continental sources and the role of different weather systems in elemental deposition. Total concentrations of trace elements were measured in aerosol extracts and wet deposition, and Se speciation was obtained with an optimized chromatographic method coupled to inductively coupled plasma tandem mass spectrometry (LC-ICP-MS/MS).

These analyses were combined with molecular organic compound analysis by pyrolysis-gas chromatography mass spectrometry (Py-GC-MS). For modelling the source contributions to Se, we used a combination of i) a Eulerian approach with the atmospheric aerosol-chemistry-climate model SOCOL-AERv2, and ii) a Lagrangian approach with air parcel backward trajectories and a moisture source diagnostics. While weekly Se measurements in the 2015-2020 aerosol time series agreed very well ($r\sim0.8$) with SOCOL-AERv2 model results, the higher Se concentrations ($>0.05$ ng·m$^{-3}$) observed in summer

were underestimated by the model. We could explain these higher concentrations in summer by convection related to thunderstorms that led to high aerosol loadings and which are not resolved explicitly in the model. In addition, convective events, associated with continental moisture sources, also explained the highest concentrations of Se and most other trace elements in wet deposition, due to efficient below cloud scavenging, indicating the importance of local cloud dynamics on the supply of Se and other, essential and non-essential, trace elements to surface environments. While data for water isotopes in

precipitation indicated an uncoupling of hydrological and trace element cycling related to below cloud scavenging, cloud water isotopes and trace elements showed high correlations indicating that the water and trace element cycles are strongly coupled from the source to the formation of clouds with a possible decoupling occurring during precipitation. Furthermore, cloud water showed more regional trace element and moisture sources than precipitation samples. With this comprehensive set of observations and model diagnostics we could explain inorganic Se speciation in unprecedented detail by linking moisture

sources and organic chemical compounds in aerosols to speciation data of Se and S, indicating local vs long-range transport and anthropogenic vs natural Se sources. We report for the first-time organic Se in precipitation (and aerosols), for which we could elucidate a marine biogenic source. Our study thus provides new insights into the factors explaining atmospheric deposition of Se and other trace elements and highlights the importance of weather system dynamics in addition to source contributions for the atmospheric supply of trace elements to surface environments.




**Graphical Abstract**

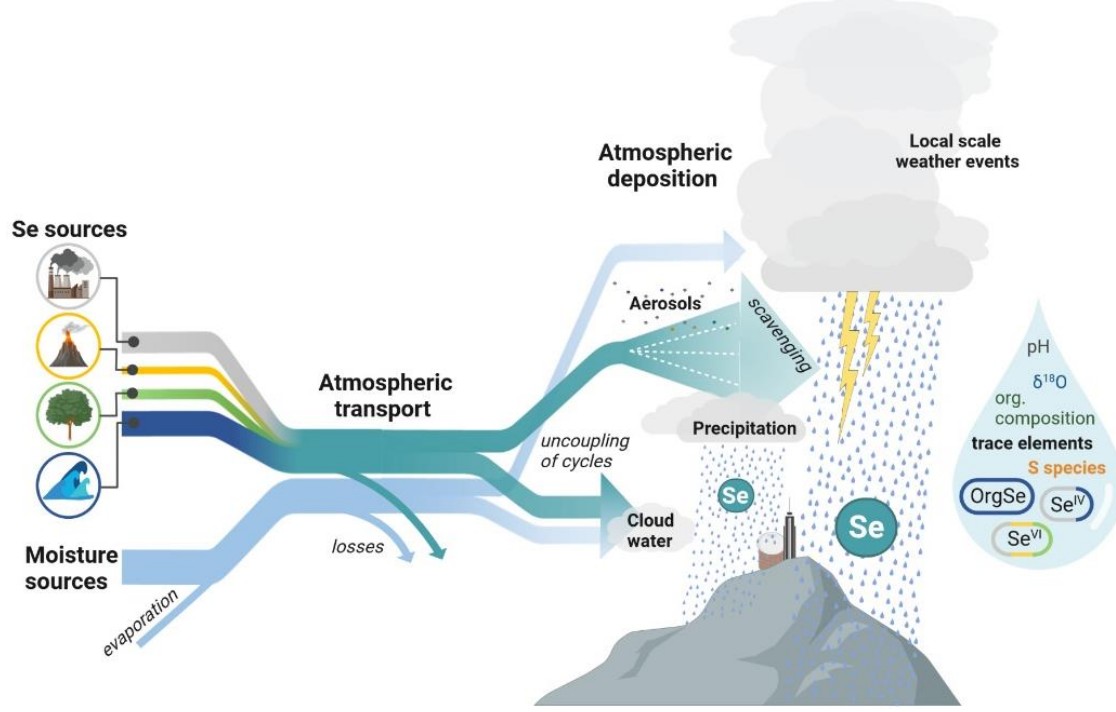



## 1 Introduction

Selenium (Se) is a key micronutrient used in the production of selenoproteins that serve essential biochemical functions in humans and many other organisms (Lobanov et al., 2009; Lazard et al., 2017). Plant-based food products are an important dietary source of Se. In turn, atmospheric deposition is an important source of Se to terrestrial ecosystems, including agricultural soils, with an estimated Se amount of 11.9-14.6 Gg being annually deposited on terrestrial environments, mostly by wet deposition (~80%) (Wen and Carignan, 2007; Feinberg et al., 2020a). Although atmospheric Se deposition affects Se levels in plant-based food products, the atmospheric Se cycle is scarcely understood. It is known that gaseous methylated selenide species, including dimethyl selenide (DMSe), can be emitted from both marine and terrestrial environments, such as wetlands (Amouroux and Donard, 1996; Amouroux et al., 2001; Vriens et al., 2014a). In contrast, volcanic degassing and anthropogenic activities have been suggested to emit inorganic Se, in the form of elemental Se ($Se^0$), selenium dioxide ($SeO_2$), hydrogen selenide, and seleno-carbonyl compounds (Yan et al., 2004; Pavageau et al., 2004). Volatile Se species emitted from both natural and anthropogenic sources are expected to quickly oxidize under atmospheric conditions, eventually forming the species $Se^0$, $SeO_2$, selenite ($Se^{IV}$) and/or selenate ($Se^{VI}$) (Wen and Carignan, 2007; Feinberg et al., 2020b), which are less volatile and likely undergo gas-particulate conversion (Wen and Carignan, 2007). Recently, Feinberg et al. (2020a; 2020b) created a global atmospheric Se model by implementing Se chemistry in an aerosol-chemistry climate model, thus updating estimates of Se emission sources: in their simulations, 65% of atmospheric Se is emitted by natural sources, including emissions from the marine and terrestrial biosphere, volcanoes, and minor contributions from sea salt and dust. The remaining 35% of total Se emissions originate from anthropogenic activities, mainly fossil fuel combustion, metal smelting, biomass burning, and manufacturing (Feinberg et al. (2020a; 2020b)). The Se model was calibrated with measurements of total Se concentration in aerosols, mostly from air quality monitoring networks, but also individual studies (Feinberg et al. (2020a) and references therein). Observations from Se in rainfall are available from limited point measurements (De Gregori et al., 2002; Wallschläger and London, 2004) and seasonal-accumulated measurements (Suzuki et al., 1981; Weller et al., 2008; Suess et al., 2019; Roulier et al., 2021). However, despite these recent estimates of emission sources and available observations, the factors driving total and Se species in atmospheric deposition remain insufficiently constrained by observations, particularly due to a lack of field measurements that separate source contributions of Se. Furthermore, atmospheric dynamics, e.g., horizontal long-range transport and local vertical transport and precipitation due to convection, can play a role in the atmospheric distribution and deposition of trace elements; however, this factor has been scarcely investigated for trace elements.

Besides knowing total Se concentrations, it is of equal importance to understand Se speciation as this dictates Se mobility in soil, uptake by plants, and ultimately health impact (Fernández-Martínez and Charlet, 2009; Winkel et al., 2015). For example, $Se^{VI}$ is highly mobile in soils and available for plant uptake, while $Se^{IV}$ is efficiently retained by sorption on oxide minerals and thus less available to plants (Ali et al., 2017). Much of the knowledge on atmospheric Se speciation has been derived from what is known for sulfur (S), a well-studied element sharing chemical properties (e.g., redox-sensitivity) and



important cycling pathways (e.g., biogenic methylation and volatilisation) with Se (Vriens et al., 2014b). Only a few studies reported Se speciation in rainfall (Cutter and Church, 1986; De Gregori et al., 2002; Suess et al., 2019; Roulier et al., 2021),

cloud water (Kagawa et al., 2022) and aerosols (Kagawa et al., 2003; De Santiago et al., 2014), which solely detected inorganic Se (predominantly $Se^{VI}$), with substantial amounts of Se remaining unidentified (59-80% species unidentified compared to total Se concentrations) (Wallschläger and London, 2004; Suess et al., 2019; Roulier et al., 2021). This limited knowledge can be attributed to the lack of analytical techniques enabling the determination of Se speciation at ultra-trace levels (sub-ng·L$^{-1}$ for precipitation and sub-pg·m$^{-3}$ for cloud water and aerosols). $Se^{IV}$ has been reported in atmospheric samples in urban

environments (Suzuki et al., 1981; De Santiago et al., 2014) and/or samples associated to anthropogenic emissions (Cutter and Church, 1986; De Gregori et al., 2002; Kagawa et al., 2003; De Santiago et al., 2014; Kagawa et al., 2022). In this context, $Se^{IV}$ and/or the ratio of $Se^{IV}$:$Se^{VI}$ in aerosols have been suggested as a tracer for coal emissions (Kagawa et al., 2003; De Santiago et al., 2014) as well as a potential redox tracer in rain and cloud water (Cutter and Church, 1986; Kagawa et al., 2022).

Although each previous study advanced the knowledge of atmospheric Se cycling, every approach used in these studies has shortcomings. Measurements give very precise information but are limited in scope and lack analysis of factors that inform on source contributions and atmospheric processing, whereas modelling offers information on a larger but coarser scale. For example, the global atmospheric Se model developed by Feinberg et al. (2020b) has a resolution of 2.8° x 2.8° and does not include specific Se species in atmospheric deposition, apart from fractions defined as "oxidized inorganic" and "oxidized

organic" Se. Therefore, the understanding of atmospheric Se cycling is fragmented, and many open questions remain, as outlined below.

Understanding the factors controlling Se speciation in atmospheric deposition, and generally atmospheric Se cycling, needs a combination of various methods targeting aerosols and wet deposition. We thus designed a methodological framework that includes different chemical measurements and modelling approaches (Fig. 1), integrating chemistry and atmospheric

dynamics to get insights into the factors controlling atmospheric Se deposition. Parameters derived from chemical analyses included total concentrations of various (trace) elements, stable water isotopes, Se and S speciation obtained via improved (ultra)sensitive methods based on liquid chromatography and inductively coupled plasma tandem mass spectrometry (LC-ICP-MS/MS), as well as organic molecular composition of aerosols based on pyrolysis-gas chromatography mass spectrometry (Py-GC-MS). These analyses are combined with modelling of Se source contributions using the atmospheric aerosol-

chemistry-climate model SOCOL-AERv2 and air parcel trajectory calculations based on the Lagrangian analysis tool LAGRANTO (Sprenger and Wernli, 2015; Wernli and Davies, 1997) with three-dimensional wind fields from the atmospheric reanalysis dataset ERA5 to estimate dominant moisture sources and atmospheric transport patterns.

We applied this framework to field sampling campaigns that we carried out at the high-altitude Pic du Midi Observatory (French Pyrenees; 2877 m a.s.l.). This site is ideally located to study long versus short-range contributions of moisture and

(trace) elements from both continental (e.g., from Western Europe and North Africa) and marine sources (Atlantic Ocean and Mediterranean Sea) (Fu et al., 2016). Furthermore, this high-altitude site enables the investigation of the effect of different





weather systems (e.g., enhanced convection or frontal passages over mountainous terrain), on trace element and moisture source cycling. We created a unique dataset of aerosols samples (n=134) taken at a weekly resolution over 5 years, as well as samples of precipitation (n=26), cloud water (n=56), and aerosol (n=48) taken at high temporal resolution (sub-event based

sampling of wet deposition or separate day and night sampling of aerosols).

Our objective in this study is to comprehensively investigate the role of different source factors and weather dynamics on the observed variability in trace element concentrations and Se speciation in atmospheric deposition. Specifically, the combined investigation of aerosol, cloud water and precipitation offers unique insights into the evolution of atmospheric deposition, by looking at effects on trace element cycling by changing air masses as well as in-cloud and below-cloud

processes. Our specific research questions were: (1) What is the contribution of different atmospheric sources to measured Se concentrations in aerosols? (2) How do different physical and chemical parameters modulate the signals of moisture and trace elements in wet deposition? (3) How do local weather events such as thunderstorms influence atmospheric deposition fluxes of Se and other trace elements? (4) What are the proportions of Se and S species in atmospheric deposition and how do these vary as a function of different chemical proxies and contributing moisture sources?




## 2    Methods

### 2.1 Sampling of atmospheric deposition

Two series of atmospheric samples were collected at the summit of the Pic du Midi Observatory monitoring station, which is located at 2877 m a.s.l. in the central Pyrenees Mountains. The first series consists of aerosol samples collected weekly
between 23rd June 2015 and 24th April 2020, and is referred to as the "2015-2020 aerosol time series". The second series consists of precipitation, cloud water and aerosols collected at high temporal resolution between 28th August and 13th October 2019, and is referred to as the "2019 campaign". An overview of the measurements and modelling approaches applied to both series is given in Fig. 1.

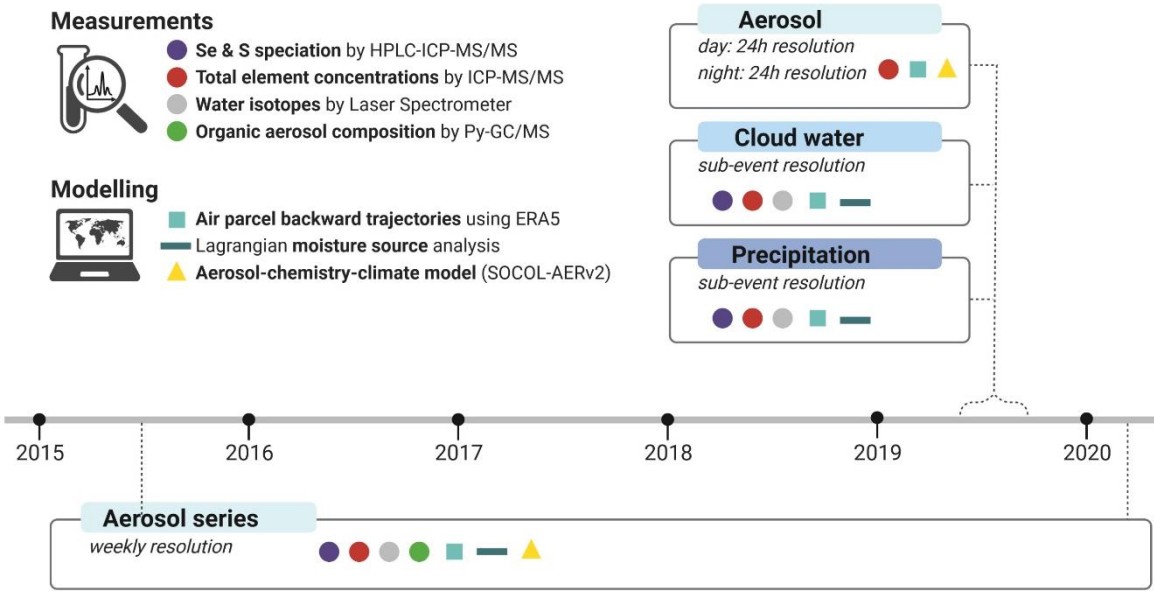

**Fig. 1.** Overview of the measurements and modelling approaches applied to the 2015-2020 aerosol time series and 2019 campaign, in which aerosol, cloud water and precipitation samples were collected. Figure created with BioRender.com.

### 2.1.1 Weekly sampling of aerosols (2015-2020)

Two different systems were used to sample aerosols during warm (summer and autumn) and cold months (winter and spring). A total of 70 weekly aerosol samples were collected from June-October between 2015 and 2019 (referred to as warm
season samples) by using a TISCH high-volume PM10 sampler located on the backside of the Bernard Lyot Telescope tower (south-west of the summit; distance of approximately 60 m from wet deposition sampling). With this system, 8×10 inch quartz filters (QM-A, Whatman) were used after muffling them at 550°C. The weekly sample collection of warm season samples was conducted between specific times, i.e. from 22:00 to 15:00 UTC between the years 2015 and 2018, and from 20:00 to 08:00 UTC during 2019. Sampling throughout the entire day was not possible due to noise interferences of the high-volume sampler
on the telescope observatory activities on the station. On average, warm season samples were collected over 7±2 days with a




collection volume of 7197±1816 m$^3$. A total of 64 aerosol samples were collected from November through May between 2016 and 2020 (referred to as cold season samples) by using 90 mm single stage Teflon filter packs (Savillex) with pre-muffled 90 mm diameter quartz filters (QM-A, Whatman). The filter packs, which were pre-cleaned before the campaign (with 10% HNO$_3$ and then ultrapure water), were connected to a Tekran 1104 Teflon-coated, heated manifold (necessary for sampling during

cold temperatures). Ambient air was pulled through the manifold at a constant flow rate of ∼100 L·min$^{-1}$ using a blower unit. Total sampling volumes were recorded using gas meters that were installed at the outlets of the vacuum pumps (KNF™ Aluminium/Chromium Diaphragm Pump, ∼55 L·min$^{-1}$). On average cold season samples were collected over 6.9±0.4 days (24 h·day$^{-1}$) with a collection volume of 518±32 m$^3$.

### 2.1.2 High temporal resolution sampling of precipitation, cloud water and aerosols (2019 campaign)

Precipitation and cloud water samples were collected on a (sub-)event basis at the aerology platform at Pic du Midi Observatory. Precipitation was sampled using 2-3 custom-made plastic (polyprolyene) collectors that were pre-cleaned (with 10% HNO$_3$ and then ultrapure water) and only opened during precipitation events (surface area per collector: 0.159 m$^2$). Cloud water samples were taken using a CASCC2 bulk fog/cloud sampler (air volume flow: 1470 m$^3$·h$^{-1}$) (Collett et al., 1990). Throughout the campaign, a total of 18 precipitation and 12 cloud water events were sampled, consisting of 26 and 58 sub-

events, respectively. Aerosol filters were separately sampled during day- and night-time over 2 days (8:00/20:00 UTC, each 24 h in total) on 47 mm single stage filter Teflon packs (Savillex, pre-burned filters: quartz QM-A Whatman) using the same system described above for the collection of cold season samples. The mean sampling volumes were 53.6±2.8 m$^3$ over a collection period of 24 h.

### 2.2 Sample processing

### 2.2.1 Aerosol samples

All aerosol filters were collected and stored in aluminium foil in the dark at -20°C. For the warm season aerosol samples, two types of extractions were performed for 1) total element analysis (using microwave assisted acid digestion), and 2) water-soluble Se and S speciation. This latter extraction was solely performed on the warm season samples due to limited sample material and lower concentrations in the cold season samples. For total element analysis (sub-samples 1), all aerosol samples

were extracted with 4 mL HNO$_3$ (69%, ROTIPURAN®supra, Roth) and 1 mL H$_2$O$_2$ (30%, for ultra-trace analysis, Sigma-Aldrich) in a microwave oven (UltraClave IV, MLS). The digestion program of Kulkarni et al. (2007) was followed, which included a first temperature increase to 180°C (15 min ramp, temperature held for 15 min) followed by a final increase to 210°C (15 min ramp, temperature held for 45 min). Acid digestions of warm season samples were done using a 3.8 cm$^2$ (filter area):25 mL$^{-1}$(digestion volume) ratio. The digestion volume of cold season samples was increased after 2017 from 5 to 15 mL

to increase available extraction volume for further analysis (final extraction ratio: 7.069 cm$^2$·15 mL$^{-1}$). For the water-soluble Se and S speciation analysis (sub-samples 2), all samples were extracted using a ratio of 11.404 cm$^2$·(filter area):15 mL$^-$





[1](ultrapure water volume). The filter:water suspensions were sonicated for 20 min at 20°C (repeated twice) and the resulting extracts were filtered (0.22 μm, 25 mm syringe filters, Nylon, BGB). 9 mL of the extracts were used for pre-concentration and subsequent Se and S speciation, while the remaining sample volume was used for total element analysis to determine extraction

efficiencies.

Furthermore, warm season aerosol filters were analysed by Py-GC-MS to identify organic compounds previously used as proxies for aerosol sources (e.g., Zhao et al. (2009)), for which 1.539 cm$^2$ of aerosol filters were punched and folded into pyrolyzer cups (Eco-cup SF, Frontier Laboratories, Japan).

**2.2.2 Precipitation and cloud water samples**

Precipitation as well as cloud water samples were processed within 2 h after collection. Snow and hail samples were thawed in closed sampling bottles before further processing. The total deposition amount of collected samples was determined by weight. At each sub-event, 4 sub-samples were taken for 1) quantification of total elements, 2) speciation of Se and S, 3) stable water isotopes and 4) pH measurement. All plastic storage containers were pre-cleaned using 1% $HNO_3$, rinsed with ultrapure water and air dried. Sub-samples 1) used for total element quantification were filtered (0.22 μm, 25 mm syringe

filters, Cellulose Acetate (CA), BGB), stabilized (1% v/v $HNO_3$ of 69%, ROTIPURAN® Supra, Roth) and stored at 4°C until analysis. Sub-samples 2) used for speciation analyses were filtered (0.22 μm, CA), immediately frozen and stored at -20°C. Sub-samples 3) used for stable water isotopes analysis were stored in closed air-tight sampling bottles until resampling into Parafilm sealed glass vials within a few hours after collection. pH measurements were done immediately after sampling with a field meter probe (Multi 3430 with pH electrode Sentix 940-3, WTW) that was calibrated daily. To enable pH measurement

of wet deposition samples, it was necessary to add a few drops of 3 mol·L$^{-1}$ KCl (Sigma Aldrich) solution to sub-samples to increase ionic strength (Thermo Fisher, 2014).

**2.3 Chemical analysis**

**2.3.1 Quantification of total element concentration**

Total element concentrations (i.e., Li, Na, Mg, Al, Si, K, P, S, Ti, V, Cr, Mn, Fe, Co, Ni, Cu, Zn, As, S, Br, Rb, Sr, Nb,

Mo, Ag, Cd, I, Cs, Ba, Pb) were quantified using an Agilent 8900 ICP-MS/MS instrument equipped with and SPS4 introduction system, and using hydrogen (H$_2$, 5 mL·min$^{-1}$, MS/MS mode), oxygen (O$_2$, 30 mL·min$^{-1}$ MS/MS mode), and helium (He, 5.5 mL·min$^{-1}$, single quad) as reaction gases depending on the element (Supplement S1, Table 1). The instrument was tuned daily before analysis with a solution containing 10 μg·L$^{-1}$ of Mg, Li, Tl, Y, Co and Ce. Instrument drift was monitored and corrected using internal standards (i.e., 1 mg·L$^{-1}$ Sc; 0.1 mg·L$^{-1}$ In, Lu; 50 μg·L$^{-1}$ Y; Supplement S1, Table 2). Instrument

performance was checked with two certified reference materials for trace elements in surface waters with dilutions of 1:10 and 1:100 (SRM NIST 1643f; TMDA 51.2, National Water Research Institute Environment Canada). Se yielded recoveries of



99±2% for a concentration range of 0.12 - 1.20 µg·L$^{-1}$. Recoveries of other (trace) elements are listed in Supplement S1, Table S3.

### 2.3.2 Sample pre-concentration and determination of Se and S speciation

The optimized pre-concentration (further details in Supplement S2) was determined to be lyophilisation of frozen samples from an initial volume of 12 mL (for precipitation) or 9 mL (for water extract of aerosol filter) to a residual volume of 1.5 mL (pre-concentration factor of 8 or 6) to which ammonium citrate was added to increase ionic strength. The initial volume of cloud water samples was variable depending on sampled amounts. Speciation of Se and S was analysed by High Pressure Liquid Chromatography (Agilent 1260 Infinity II Bio-inert HPLC System) coupled to either an 8900 (Se speciation) or Agilent

8800 (S speciation) ICP-MS/MS. We optimized the chromatographic separations of Se species based on the method published by Tolu et al. (2011). The detailed optimizations of the chromatographic separation of Se are described in detail in Supplement S3, including specific HPLC-ICP-MS/MS operating conditions for speciation analysis of Se and S (Supplement S3, Tables S5-6). The optimized chromatographic method includes separation of Se species by anion exchange using a PRPX-100 column (Hamilton, 2.1 x 150 mm, 5 µm) equipped with an in-line filter (Titanium Frit 0.5 µm, 10-32 Waters type, BGB). Se separation

was done by gradient elution with ammonium citrate (5.2-13 mM, 2% MeOH, pH 5.2) delivered at 0.5 mL·min$^{-1}$ and an injection volume of 20 µL. For S, the chromatographic separation method developed by Müller et al. (2019) was applied, which includes gradient elution with formic acid (from 24 to 240 mM) delivered at 1 mL·min$^{-1}$ and an injection volume of 50 µL, using a Hypercarb column (100x 4.6 mm, 5 µm, Thermofisher) equipped with guard column. For Se species detection, the ICP-MS/MS operated in MS/MS mode with $H_2$ as a reaction gas (5 mL·min$^{-1}$) and an acquisition time of 100 ms for all Se

isotopes: ($m/z$ 74→74, 76→76, 77→77, 78→78, 80→80 and 82→82). To check for potential interferences, Br was monitored during all Se analyses (acquisition time: 50 ms, $m/z$ 79→79, 81→81). For S species detection, the ICP-MS/MS operated in MS/MS mode with a mixture of $O_2$ (30%) and $H_2$ (1 mL·min$^{-1}$) and an acquisition time of 50 ms for S ($m/z$ 32→48). To monitor changes in sensitivity, a solution containing 30 µg·L$^{-1}$ of Y (acquisition time: 10 ms, $m/z$ 89→89) and 14% v/v tetramethylammonium hydroxide was continuously supplied post-column by using the peristaltic pump of the ICP-MS/MS.

Both Se and S speciation analyses were measured in duplicates (i.e., two injections per sample).

All stock solutions of Se and S were prepared by weighing analytical grade reagents and ultrapure water (18.2 mΩ·cm Thermo Fisher, Barnstead NANOpure DIamond) in acid-cleaned vials and stored in the dark at 4°C. The following standards were used: Se: selenite (Se$^{IV}$, 99%) and selenate (Se$^{VI}$, 99.99%), *D,L*-selenomethionine (SeMet, ≥99%), seleno-*L*-cystine (SeCys$_2$, 95%); S: dimethyl sulfoxide (DMSO, ≥99.9%), dimethyl sulfone (DMSO$_2$, ≥99%), methanesulfonic acid (MSA,

≥99.5%), methanesulfinic acid (MSIA, 85%), sodium formaldehyde bisulfite (i.e., hydroxymethanesulfonate: HMS, 95%) and sulfate (SO$_4^{2-}$, 99%). All standards were purchased from Sigma Aldrich, except for SO$_4^{2-}$ and DMSO$_2$, which were purchased from Merck. Stock solutions of SeCys$_2$ were dissolved in 0.2% HCl for stabilization purposes. Working standard solutions were prepared on the day of analysis by dilution in ultrapure water.




Limits of detection (LODs) were calculated according to IUPAC recommendations, i.e., the LOD equals three times the

Limits of detection (LODs) were calculated according to IUPAC recommendations, i.e., the LOD equals three times the
standard deviation of the blank baseline signal divided by the calibration slope based on peak height.

### 2.3.3 Determination of organic compounds in aerosol samples

The analysis of organic compounds in aerosols was performed with a FrontierLab pyrolyzer equipped with a FrontierLab
AS-1020E autosampler and connected to a Thermoscientific Trace 1310 GC coupled to a Thermoscientific ISQ 7000 MS. The
operating conditions and subsequent data processing method were done according to Tolu et al. (2015) (see detailed description
in Supplement S4).

### 2.3.4 Analysis of stable water isotopes

Stable water isotopes in precipitation and cloud water samples were measured using a wavelength-scanned cavity ring-
down spectrometer (L2130-i, Picarro Inc., Santa Clara, California, USA). Measurements and calibration was done following
the protocol by von Freyberg et al. (2022). Briefly, samples were injected six times, from which the first three were discarded
to take into account potential memory effects. Measurements were calibrated to the VSMOW2-SLAP scale as described by
the International Atomic Energy Agency (IAEA, 2017). The isotopic abundances of $^{18}$O and $^2$H are reported using the δ
notation relative to the IAEA standard Vienna Standard Ocean Water 2 (VSMOW2) as described in Gat and Gonfiantini
(1981).

### 2.4 Data and modelling

### 2.4.1 Particle number, black carbon and observational meteorological data

A wide range of meteorological and other parameters characterizing the physical properties and chemical composition of
the atmosphere at Pic du Midi Observatory are routinely monitored and are available through an internet database
(https://p2oa.aeris-data.fr). For this study, we used equivalent black carbon (BC) mass concentrations that were analysed by
an Aethalometer (Magee Scientific Co. AE 33) with a time resolution of 15 min as described in Hulin et al. (2019). The particle
number (PN) was analysed by a condensation particle counter (TSI, Inc., 3010, 3750) with a time resolution of 5 min.
Considered meteorological data included air temperature (Vaisala QMH 101), relative humidity and atmospheric pressure
(both measured with Vaisala MAWS 301), which were all extracted with a time resolution of 60 min.

### 2.4.2 Air parcel back trajectories and moisture source analysis

To estimate dominant moisture sources of precipitation and atmospheric transport patterns, trajectories were calculated
with the Lagrangian analysis tool LAGRANTO (Sprenger and Wernli, 2015; Wernli and Davies, 1997) with three-dimensional
wind fields from the atmospheric reanalysis dataset ERA5 (Hersbach et al., 2020) interpolated to a 0.5°x0.5° horizontal grid
on 137 vertical levels. The air parcels' hourly position (longitude, latitude, pressure) are calculated for 7 d backward in time





from a set of five starting points including the location of the Pic du Midi Observatory (42°56'11.4"N 0°08'31.1"E) as well as four other points shifted by 0.5° in the horizontal to take into account uncertainties in the origin of air at the synoptic scale due to turbulent mixing in the boundary layer and the impact of orography. A set of variables is interpolated along the trajectories, namely, specific humidity, relative humidity, as well as the boundary layer height. Trajectories were started at the surface to assist the interpretation of the aerosol data, as well as from a vertical stack of 9 points every ~80 hPa between 900 hPa and 300 hPa for assessing the origin of precipitating air. The trajectories were started every hour for the 2019 campaign period, and every 6 h for the climatological period between 2015 and 2020 of the aerosol time series.

The moisture sources of precipitation and cloud water were calculated using the algorithm developed by Sodemann et al. (2008). In short, the mass budget of water vapour in an air parcel is considered and moisture uptakes are registered whenever the specific humidity along an air parcel trajectory increases. The weight of each uptake depends on its contribution to the specific humidity of the trajectory upon arrival. If precipitation occurs underway (manifesting itself by a decrease in specific humidity along the trajectory) after one or several uptakes, the weight of all previous uptakes is reduced proportionally to their respective contribution to the loss.

The position and intensity of the precipitation systems in ERA5 are not always placed at the correct geographical location, in particular if they are of convective nature. For example, no precipitating cloud may be present in ERA5 at Pic du Midi when precipitation was actually sampled. Therefore, we used a slightly different approach than Sodemann et al. (2008) to select precipitation and cloud water events. We based our event selection criteria on the actual precipitation and cloud water sampling and diagnose the origin of water vapour during the respective sampling periods (similar to the approach in Aemisegger et al. (2014); Pfahl and Wernli (2008)). For precipitation we use all available trajectories in the vertical column for which the relative humidity at arrival is close to saturation (>80%) during the sampling time interval and for the cloud water samples we use near-surface water vapour. In both cases, we weight the contribution of individual moisture uptakes along each trajectory according to its specific humidity at arrival.

### 2.4.3 Chlorophyll-*a* exposure in marine moisture source regions

To investigate a potential link between marine productivity and Se (speciation) in rainfall, chlorophyll-*a* (Chl-*a*) air exposure in marine moisture source regions was diagnosed with the same approach as in Suess et al. (2019). Sea surface water Chl-*a* concentrations from the Ocean Colour Climate Change Initiative dataset, version 3.1 from the European Space Agency (available online at http://www.esa-oceancolour-cci.org/) were used to estimate the air parcels' exposure to Chl-*a* in marine moisture source regions. The chosen CHL-OC5 Chl-*a* product (0.01°x 0.015° gridded data) is based on daily remote-sensing reflectance data of MODIS/Aqua and MERIS/ENVISAT.16,17. To minimize the impact of regions with missing data due to cloud cover in all seasons, two-weekly composites were computed to estimate the time mean moisture source Chl-*a* exposure. A similar approach to determine air exposure to marine Chl-*a* was used in Blazina et al. (2017), with the difference that the exposure of the air was computed based on trajectory points located in the boundary layer. Here, we focus on Chl-*a* exposure





305 of air at instances when they have taken up humidity along their paths. We thus use humidity uptake along the trajectories as an analogue for enhanced uptakes of biogenically derived compounds from the ocean to the atmosphere.

**2.4.4 Data sources for thunderstorms**

Lightning data was used to detect thunderstorm activity in the vicinity of Pic du Midi Observatory. Two-hourly lightning maps and data from the nearest radio antenna stations were extracted from the Meteo60 website 310 (https://www.meteo60.fr/orages-archives/, accessed 24th November 2022), which take their data from the Blitzortung network (list of stations: https://www.blitzortung.org/en/station_list, accessed 24th November, 2022). Atmospheric deposition samples were classified as thunderstorm events, if lightning activity was reported in the vicinity of Pic du Midi Observatory (within 3 km) during the time that a sample was collected. The vicinity was defined by comparing detected lightning with on-site observations during the 2019 campaign.

315 **2.4.5 Atmospheric Se modelling with SOCOL-AERv2**

We conducted 2015–2020 simulations in the aerosol-chemistry-climate model SOCOL-AERv2, which includes Se chemistry (Feinberg et al., 2019; Feinberg et al., 2020b). Selenium cycling is modelled with 7 gas phase species, 40 particulate tracers in size bins between 0.39 nm and 3.2 µm, and 17 chemical reactions (Feinberg et al., 2020b). Global simulations are conducted at T42 horizontal resolution (2.8° × 2.8°) with 39 vertical levels up to 80 km. Model dynamical variables (vorticity 320 and divergence of the wind fields, temperature, and surface pressure) are nudged towards ERA5 reanalysis data (Hersbach et al., 2019) in order to minimized the differences between observed and modelled meteorology in these simulations. Emissions of Se from anthropogenic, marine biogenic, terrestrial biogenic, and volcanic sources are considered, with emission totals constrained by a previous study applying Bayesian inference methods with available Se observations (Feinberg et al., 2020a). The SOCOL-AERv2 Se simulation has been thoroughly validated against available particulate and wet deposition Se 325 measurements (Feinberg et al., 2020a; Feinberg et al., 2020b; Feinberg et al., 2021). The spatial distribution and trend in anthropogenic Se emissions are based on the CAMS-GLOB-ANT-v4.2 $SO_2$ emissions inventory for 2015–2020 (Granier et al., 2019). The spatial distribution of marine biogenic emissions is calculated online by applying a wind-driven parametrization (Nightingale et al., 2000) to a marine DMS climatology (Lana et al., 2011). Terrestrial biogenic emissions of Se are distributed following the mean monthly emissions of volatile organic carbon (VOC) emissions from the MEGAN-MACC inventory 330 (Feinberg et al., 2020a; Sindelarova et al., 2014). Volcanic degassing emissions of Se are temporally constant and distributed according to the magnitude of volcanic $SO_2$ emissions (Andres and Kasgnoc, 1998; Dentener et al., 2006). Selenium is removed from the atmosphere through wet and dry deposition parametrizations which depend on the calculated grid cell meteorology. Source tracking simulations were conducted by turning on only one of the four Se emission sources (anthropogenic, marine, terrestrial, or volcanic). Modelled particulate Se concentrations at the grid cell and altitude of Pic du Midi were extracted and 335 compared to measured quantities.





## 2.5 Statistical analysis

All statistical analyses were performed using IBM SPSS Statistics 26. Principal component analysis (PCA) was performed on the precipitation data set given in mass per volume ($g \cdot L^{-1}$) or per area ($g \cdot m^{-2}$). Prior to the PCA, the data was converted to Z-scores (average=0, variance=1). Principal components (PCs) with eigenvalues > 1 were selected and extracted using a

340    Varimax rotated solution. A significance threshold value of > 0.39 was chosen. Other variables, e.g., precipitation volume, black carbon and total column ice cloud water contents and moisture sources, were included passively in the PC-loading plots by using bivariate correlation coefficients between these variables and the PC scores of each PC. Correlations were performed as Spearman rank correlations (correlation coefficient indicated by $r_S$). Significance levels (2-tailed) of performed correlations are indicated by $p<0.05$ or $p<0.01$.



# 3    Results and Discussion

## 3.1 Factors driving total Se deposition fluxes

### 3.1.1 Variability in total Se concentrations and source contribution of Se in aerosols

In a first step, we investigated Se concentrations in aerosols taken at weekly resolution over 5 years (2015-2020) to get insights into the synoptic timescale variability of total Se deposition at Pic du Midi Observatory and how this is affected by different sources. Total Se concentrations (after digestion using $HNO_3$ and $H_2O_2$) ranged from 0.002 to 0.221 ng·m$^{-3}$ in the 2015-2020 aerosol time series (n=134; average: 0.058±0.057 ng·m$^{-3}$; Fig. 2 and in Supplement S5, Fig. S4). We clearly see higher Se concentrations in summer (June-August; 0.114±0.050 ng·m$^{-3}$) than during spring (March-May; 0.032±0.024 ng·m$^{-3}$), autumn (September-November; 0.049±0.044 ng·m$^{-3}$) and winter (December-February; 0.009±0.005 pg·m$^{-3}$). Similar seasonal differences, with highest Se levels in summer, was also observed in an aerosol time series collected at the Neumayer station in Antarctica (Weller et al., 2008), for which the authors suggested that this seasonal trend is driven by biogenic Se emissions. In our study, we compared Se concentrations to many other (trace) elements and most of them show a similar pattern with higher concentrations in summer. This is interesting as it is likely that these elements are derived from different sources, e.g., mineral dust (e.g., Fe and Mn) or anthropogenic activities (e.g., Pb, Supplement S5, Fig. S5), which would indicate that elemental concentration patterns are not only driven by sources but also other (physical) processes. Higher aerosol loadings are likely related to direct exports from the boundary layer through convection or by anabatic (i.e. upslope or valley) wind systems (Hulin et al., 2019).

To determine the main Se source contributions of the 2015-2020 aerosol time series, we used the aerosol-chemistry-climate model SOCOL-AERv2. It should be noted that model grid cells represent an area of ~300×300 km$^2$, which is much larger than our sampling site and thus give regional averaged rather than site-specific information. The SOCOL-AERv2 model identifies four major Se sources: anthropogenic, volcanic, marine, and terrestrial sources (Fig. 2). We found that proportions of terrestrial contributions are highest in summer with 41±7% (of total estimated sources) followed by autumn (21±10%), spring and winter (both: 9±4%), as can be expected from a previous study conducted at Pic du Midi (Suess et al., 2019). In contrast, marine sources are relatively more important in spring (53±20%) and winter (48±14%) compared to autumn (34±15%) and summer (24±7%). Anthropogenic and volcanic sources (for our study site this is mainly the Etna volcano in Sicily) do not show clear seasonal variations, with average contributions of 32±8% and 8±10%, respectively.



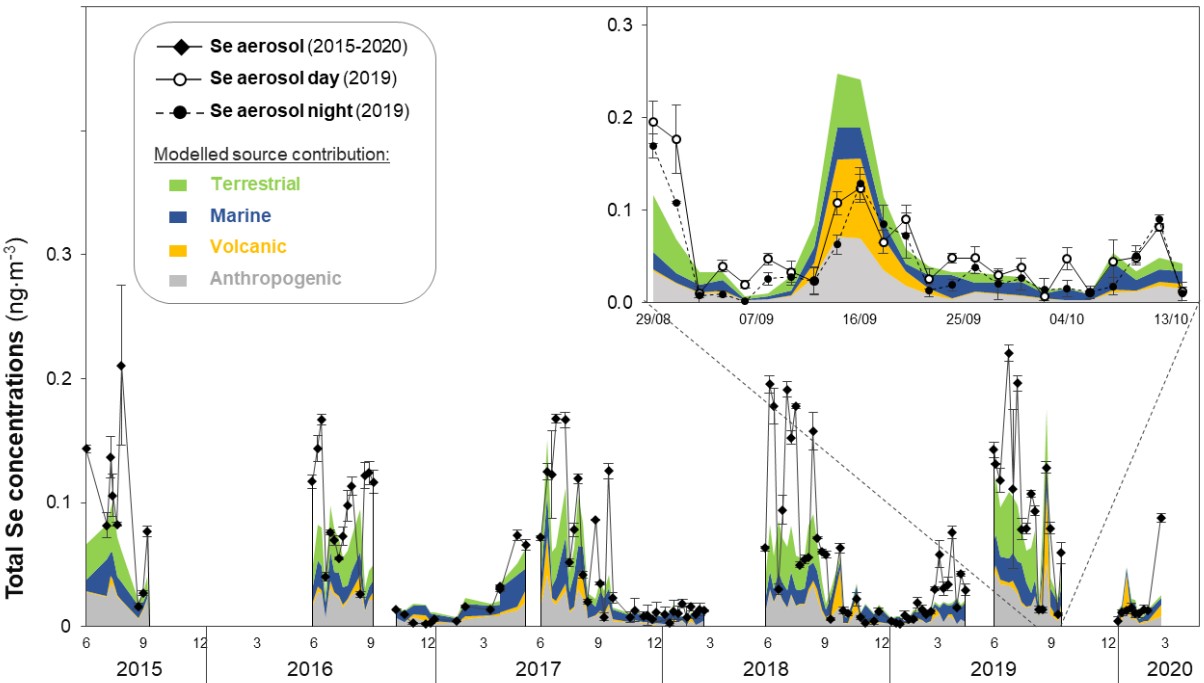

**Fig. 2.** Comparison of Se aerosol measurement with modelled Se data. Data shown for 2015-2020 aerosol time series (filled diamond) as well as day (outlined circles with continuous line) and night (filled circles with dashed line) aerosol measurements in 2019 campaign (campaign period shown in top right corner). Modelled source contributions by SOCOL-AERv2 are shown as stacked areas, including biogenic terrestrial (green), marine (blue), volcanic (yellow) and anthropogenic activities (grey).

Overall, the agreement between the weekly Se measurements in the 2015-2020 aerosol time series and the model is high ($r_S$=0.822; Supplement S6, Fig. S6). A previous SOCOL-AERv2 evaluation found similar correlation values ($r$~0.8) in a spatial comparison with annual means from measurement stations worldwide (Feinberg et al., 2020a). Our comparison here with higher time resolution measurements acts as an additional test of the model's ability to capture the temporal variability of Se sources and transport. The overall good correlation provides confidence in the model's predictions of Se source contributions at Pic du Midi Observatory. For example, lower Se concentrations (<0.05 ng·m$^{-3}$) are well captured by the model and are predominantly characterized by marine source contributions (Fig. 2 and in Supplement S6, Fig. S7b). However, the model displays a negative bias (-0.016 ng·m$^{-3}$), underestimating the higher Se concentrations (>0.05 ng·m$^{-3}$) observed in summer (Fig. 2). Such a bias is to be expected when comparing a relatively coarse global model simulation with observations from a mountain station. At the horizontal grid spacing of 300 km the model does not capture the complex terrain and associated boundary layer dynamics around Pic du Midi that lead to enhanced boundary layer contributions and higher aerosol loadings. In addition, a model grid point value always represents an area average, in contrast to point measurements. Particularly terrestrial source contributions seem to be underestimated (Supplement S6, Fig. S7a), whereas volcanic source contributions tend to be overestimated by the model (particularly during the 2019 campaign period in Fig. 2; and Supplement S6, Fig. S7c). This discrepancy is likely related to the (spatially and temporally) dynamic nature of volcanic emissions. So far, the model



only considers constant Se degassing emissions from volcanoes with a globally uniform Se:SO₂ emission ratio, due to very limited available emission data (Feinberg et al., 2020a).

The aerosol data are thus useful to understand seasonal variability in Se concentrations, however, Se deposition to the surface occurs predominantly via wet deposition (~80%) (Wen and Carignan, 2007; Feinberg et al., 2020a), and therefore it is
also important to further understand what drives concentrations in wet deposition.

### 3.1.2 Variability of elemental concentrations and moisture sources in wet deposition

During the campaign in 2019, total Se concentrations in precipitation ranged from 12 to 84 ng·L⁻¹, resulting in total Se deposition of 11 to 428 ng·m⁻² (n=26, average 42±21 ng·L⁻¹ and 85±92 ng·m⁻², respectively; Supplement S5, Fig. S4). It should be noted that concentrations in aerosols and wet deposition cannot be directly compared because they are given in
different units. These Se concentrations are similar to ranges reported for precipitation at remote and/or continental sites (6-85 ng·L⁻¹; Suess et al. (2019); Roulier et al. (2021)) and the open ocean (20-40 ng·L⁻¹; Pacific: Arimoto et al. (1985), Atlantic Ocean: Cutter (1993); Cutter and Church (1986)). They are, however, much lower than the Se concentrations found at coastal (310-2830 ng·L⁻¹; Blazina et al. (2017)) and urban sites (50-1010 ng·L⁻¹; Tokyo Japan: Suzuki et al. (1981), Sri Lanka : Savage et al. (2017)). Selenium concentrations in cloud water ranged between 0.018 and 0.967 μg·L⁻¹, corresponding to 1-16 pg·m⁻³
(n=56 samples, 5±4 pg·m⁻³; Supplement S5, Fig. S4). To the best of our knowledge, there are no previous studies on Se concentrations in cloud water in Europe. Only a small number of samples have been collected at mountain sites in Japan (Kagawa et al., 2022), Pakistan (Ghauri et al., 2001) and the USA (Richter et al. (1998); Yang and Husain (2006)), with higher Se concentrations (0.2-9.2 μg·L⁻¹) that the authors related to anthropogenic sources.

To investigate potential parallels between trace element cycling and the atmospheric water cycle, we studied the variability
of elemental concentration in wet deposition under different contributing moisture sources and used water isotope signals as indicators of precipitation formation conditions. Precipitation and cloud water samples collected at Pic du Midi were influenced by both land and oceanic moistures sources with approximately equal contributions for both sample types (Supplement S7, Fig. S8). Most precipitation samples were characterized by high contributions of more remote moisture sources coming from the North Atlantic (Precipitation events: P2-5, P13-18, average moisture contribution: 58±17% versus 8±7% in other events)
or North Africa (P6-P10, average moisture contribution: 26±9% versus 1±1% in other events). In contrast, most cloud water samples were predominantly influenced by regional sources (Spain north of 41.8°N and France south of 43.6°N, C1-3, C8-10; average moisture contribution: 51±12% versus 34±15% in other events), with remaining events showing high contributions from the North Atlantic (C4, 9, 11, average moisture contribution: 69±8% versus 35±19% in other events). The isotope signatures in precipitation (δ²H=-45.4±15.6‰ and δ¹⁸O=-7.6±1.9‰) showed a larger and more depleted range than cloud
water (δ²H=-26.7±13.4‰ and δ¹⁸O=-5.3±1.6‰), which is likely due to higher precipitation formation altitudes with the contributing air being more depleted due to the rainout effect with increasing height (Dansgaard, 1964). In cloud water, we observed a positive correlation between water isotopes and concentrations of Se ($p<0.01$) as well as with almost all analysed (trace) elements, except for Si (δ²H: p=0.054) and Fe (δ²H: p=0.046), which are mineral dust indicators. This correlation,





which seems to be mainly arising from cloud water events with regional moisture sources (Supplement S7, Fig. S9), was not

observed for precipitation (Supplement S7, Fig. S10). The link between (trace) elements and water isotopes in cloud water is likely explained by the predominantly regional/local sources (in a 0.5° radius around Pic du Midi Observatory) of both trace elements and atmospheric water, except for mineral dust (Si, Fe) that is derived from more long-distance sources. In the case of precipitation, low $\delta^2$H and $\delta^{18}$O indicate a loss in heavy water isotopes during long-range transport (e.g., due to rainout or mixing of air underway). Below-cloud processes may further alter trace element and water isotope signals in precipitation,

including (i) below cloud scavenging of aerosols containing trace elements, and (ii) rain droplet evaporation and equilibration, which may lead to a pre-concentration of elements and complex alteration of the isotopic signatures. These processes lead to an uncoupling of transport pathways for stable water isotopes and (trace) elements in precipitation in contrast to cloud water.

### 3.1.3 High Se and other elements deposition associated with deep convective activity during thunderstorms

To get a better insight into processes affecting Se concentrations in wet deposition, we analysed chemical and physical

characteristics of precipitation events sampled during the campaign in 2019. During this campaign, different (mixed-) precipitation types including rain, hail, snow, sleet, light rain and/or cloud water (fog) were collected with precipitation rates ranging between 0.1 and 10.2 mm·h⁻¹; the highest rates occurred during events that included hail (Supplement S7, Table S8). Variability in both elemental concentrations in precipitation (given in µg·L⁻¹) and deposition fluxes (given in µg·m⁻²) were explored using principal component analysis (PCA). Elemental concentrations in precipitation are expected to be influenced

by the dilution effect (Gatz and Nelson Dingle, 1971), which predicts lower concentrations for increasing rain volumes. However, the role of different precipitation types on this effect has not been investigated before, to the best of our knowledge. Our data suggests that for events with light rain and/or cloud water, concentrations of the majority of analysed (trace) elements are significantly affected by precipitation volume, while no effect was observed for other precipitation types (PCA description in Supplement S8.1).

Element deposition fluxes (in µg·m⁻², not influenced by the dilution effect) were generally high between the 14th and 19th September 2019 with two characteristic maxima for most (trace) elements (events P8 and P11.1, highlighted by a vertical line in Fig. 3a). The precipitation events with high elemental deposition (P6-P11) plot on the positive side of the first principal component (PC1: 42% of total variance; Fig. 3b), indicating similar source or process factors. A positive correlation to PC1 groupings was observed for moisture sources coming from North Africa ($p<0.01$, Fig. 3b) as well as the western

Mediterranean, which is expected as it follows the same trajectory from Africa to Pic du Midi. During this period we visually observed a brownish-red colour of collected samples (i.e., precipitation P7-10, cloud water C5, and aerosols A9-11) as well as higher pH values in precipitation (pH 6.2-7.2 in P8-10 versus pH 5.4-5.8 in all other events), which is a known effect of dust deposition during so called "red-rains" (Chester et al., 1996), indicating increased dust loadings. Furthermore, these events were characterized by significantly higher contents of black carbon (BC, Mann-Whitney-U test, $p<0.01$) and total column ice

water contents (TCIW given in mm, integrated over the troposphere from the ERA 5 reanalysis dataset; Supplement S8.2, Fig. S12), which both positively correlate with PC1 groupings ($p<0.01$; Fig. 3b). Both mineral dust and BC aerosols are known to





play an important role in heterogeneous nucleation by acting as effective ice nucleating particles (Kanji et al., 2017). Apart from the presence of dust, BC, and TCIW, precipitation events with high elemental deposition in this period were associated to thunderstorms (thunderstorms identified both from visual observations during sampling and from lightning data of the Blitzortung's network; Mann-Whitney-U test, $p<0.01$; Supplement S8.2, Fig. S13). PC scores of sub-events classified by their precipitation type, including events with rainwater/hail, light rain/cloud water, snow/sleet or thunderstorms, show groupings on PC1 and PC2. Particularly sub-events with thunderstorms show positive PC scores on PC1, which indicates a significant influence of this factor on the element grouping of PC1 (Fig. 3c). Thunderstorms were predominantly related to continental moisture sources (70% versus 46% in non-thunderstorms) and much less to Atlantic moisture sources (7% versus 47% in non-thunderstorms). With respect to the higher elemental contents in wet deposition, thunderstorm clouds extend to the upper troposphere and are generally characterized by higher rainfall rates and larger rain droplets (Tost et al., 2006) leading to efficient below-cloud scavenging of aerosols. A previous study observed high mercury (Hg) concentrations in wet deposition during thunderstorms in the eastern United States, which the authors explained by scavenging of gaseous $Hg^{II}$ and $Hg^{II}$ aerosols in updrafts and downward mixing of tropospheric Hg (as $Hg^{II}$) to lower altitudes (Holmes et al., 2016).







**Fig. 3.** Precipitation chemistry and (sub-)event characteristics. **(a)** Variability in concentrations of all measured elements in precipitation collected during the 2019 campaign. **(b)** Loading and **(c)** score plots for PCs 1–2 of the elemental deposition data set. Four PCs were retained, i.e., PC1: 42%, PC2: 26%, PC3: 25%, PC4: 2% of total variance. For the PC loadings, black circles correspond to active variables, the variables black carbon (BC, brown diamond), total column ice water contents (TCIW, blue triangle) as well as moisture sources (including Atlantic, America, Africa and Western Med. Sea, shown as filled squares) were added passively ($p<0.01$). Significance level of >0.39 is indicated by a horizontal and vertical dashed line in the loading plot. Additional information on precipitation events in (a, c) include: observed dust colour in samples (brown bar), thunderstorms (lightning bolts), and precipitation type (rain/hail: blue squares, snow/sleet: light grey circles; light rain/ cloud water: dark grey triangles). Two significant elemental deposition increases discussed in the text are highlighted by dashed line in (a).

In a next step, we investigated the importance of thunderstorms for elemental concentration trends in aerosols in the 2015-2020 time series. Also for this dataset, elemental concentrations in aerosols were significantly higher during weeks in which thunderstorms occurred than for samples that were not related to thunderstorms (selection of major and trace elements in Fig.





4; Mann-Whitney-U test, *p*<0.01; data for all analysed elements in Supplement S8.3). For Se, these highest concentrations were generally underestimated in the SOCOL-AERv2 model (Supplement S8.3, Fig. S16). Significantly higher elemental

concentrations associated with thunderstorms were not only observed when considering the full aerosol time series, but also when only considering measurements in summer (Mann-Whitney-U test, *p*<0.05 except *p*=0.06 for Ni and *p*>0.2 for Ti, Zn and Mo; Supplement S8.3, Table S9), during which average concentrations are generally higher. Higher elemental concentrations are a clear indicator of higher aerosol loadings. Previous studies suggested that high aerosol loading may trigger a chain of processes that ultimately increase the strength of convective updrafts. Related to this, higher frequencies of lightning

has been linked to high aerosol loadings (Williams and Stanfill, 2002; Zhao et al., 2015; Thornton et al., 2017; Pan et al., 2022). Within the cloud and updrafts, condensed water and ice can efficiently scavenge soluble gases and aerosols containing elements (as previously observed for Hg, Holmes et al. (2016)). Notably, we found a positive correlation between the occurrence of thunderstorms and the absolute humidity at Pic du Midi (Supplement S8.3, Fig. S17). Higher humidity likely favours stronger convection and finally wash-out of aerosols via precipitation.


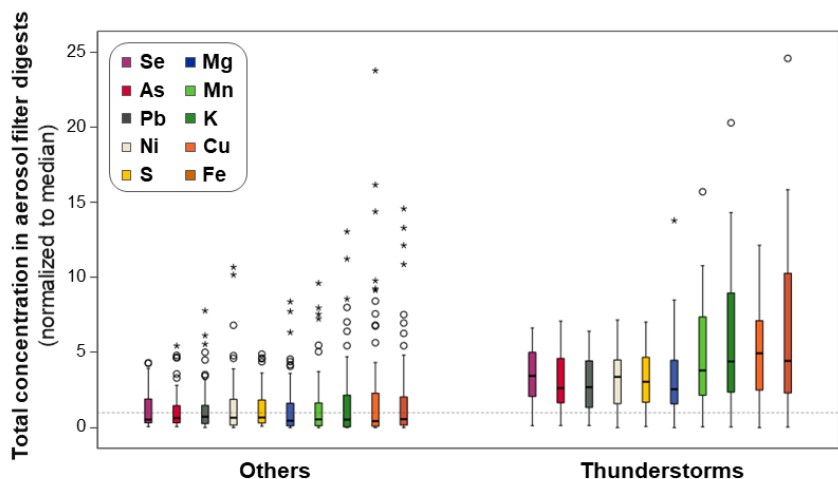

**Fig. 4.** Comparison between normalized total concentrations of different major and trace elements in aerosol filter digests sampled during weeks with thunderstorms (right side). All other aerosol measurements without detected thunderstorms during sampling are shown on the left ("others"). Shown total concentrations were normalized by median of respective element (values equal to median indicated as a horizontal
dashed line; y=1). All shown aerosols measurements during weeks with thunderstorms are significantly different (Mann-Whitney-U test; *p*<0.01).

Apart from the finding that deep convective activity during thunderstorms lead to significantly higher elemental deposition, thereby impacting supply of micronutrients and potentially toxic elements to the surface, our results imply that high elemental concentrations do not necessarily indicate a source signal in deposition data. More specifically, our data

suggests that while element concentrations in the atmosphere may be primarily determined by the element fluxes at the source, local cloud dynamics of different weather events control the amounts of Se and other (trace) elements in atmospheric deposition



fluxes. This implies that variability in elemental concentrations in wet deposition does not reflect changes in strengths of emission sources alone but also weather conditions during atmospheric removal.

## 3.2 Factors driving the deposited chemical form of Se

We further investigated the chemical forms (speciation) of Se in atmospheric samples and compared these to speciation data for S to further investigate potential source signatures. Furthermore, speciation in atmospheric deposition is relevant for the bioavailability of Se delivered via rainfall to soils and crops, as elemental speciation is an important factor for plant-uptake and mobility in soils.

### 3.2.1 Detection of inorganic and organic Se species

To determine Se speciation in atmospheric samples at ultra-trace levels, we developed (1) an extraction method for the water-soluble fraction of aerosols, (2) a pre-concentration method based on volume reduction by lyophilisation, which was applied to aerosol water extracts, precipitation and cloud water, and (3) an optimized a LC-ICP-MS/MS method (detailed procedure in method section and in Supplements S2-3). Our optimized procedure reaches a detection limit of 1-2 $ng \cdot L^{-1}$ depending on Se species, which is 5 to 10 times lower than existing methods (Suess et al., 2019; Roulier et al., 2021). Thus,

we could detect not only the oxyanions $Se^{IV}$ and $Se^{VI}$ in all sample types, but also a third Se peak that is likely of organic nature, here further defined as OrgSe (see chromatograms in Supplement S9, Fig. S18a). Further analysis using a cation exchange column showed several unknown peaks, with one found to co-elute with an in-house synthesized standard of the metabolite dimethylselenonium propionate (DMSeP), which could indicate that this peak consists of small methylated Se compound(s) (Supplement S9, Fig. S19). With our method, Se species recoveries accounted for 65±13% of total Se

concentrations in aerosol water extracts, 77±14% in cloud water, and 66±10% in precipitation (Supplement S9, Fig. S20). Previous studies solely detected inorganic Se, mostly $Se^{VI}$ ($Se^{IV}$ being below detection limits in 89-96% of samples) and had much lower Se species recoveries (20-42%) (Suess et al., 2019; Roulier et al., 2021). To achieve full identification of the organic Se species, high-resolution mass spectrometry may further be used given that sufficient pre-concentration can be reached.

Average concentrations of $Se^{VI}$, $Se^{IV}$ and OrgSe accounted for 43.7±27.6 $pg \cdot m^{-3}$, 5.1±4.0 $pg \cdot m^{-3}$ and 2.5±1.7 $pg \cdot m^{-3}$ in aerosol water extracts; 2.7±2.3 $pg \cdot m^{-3}$, 0.7±0.8 $pg \cdot m^{-3}$, and 0.3±0.2 $pg \cdot m^{-3}$ in cloud water; and 43.6±55.3 $ng \cdot m^{-2}$, 10.2±9.9 $ng \cdot m^{-2}$ and 1.9±2.1 $ng \cdot m^{-2}$ in precipitation, respectively. We investigated proportions of Se species rather than absolute concentrations for further data analysis to allow comparison between different atmospheric samples, since the use of species proportions removes influences by e.g., variability of aerosol loading or precipitation volume (dilution effect) among

specific seasons and events. The main Se species in all analysed atmospheric samples is $Se^{VI}$, followed by $Se^{IV}$ and OrgSe (Fig. 5). $Se^{VI}$ accounts for 53.4±12.9 (aerosol), 48.7±12.8 (precipitation) and 62.9±14.0 (cloud water) % of total Se concentrations, while $Se^{IV}$ represents 8.2±8.3 (aerosol), 13.3±11.8 (precipitation) and 15.2±9.1 (cloud water) % of total Se concentrations. OrgSe accounts for the smallest fraction with 3.2±1.6 (aerosol), 3.9±2.4 (cloud water) and 2.2±1.2 (precipitation) % of total





Se concentrations (Fig. 5). Besides Se species, we detected the S species sulfate ($SO_4^{2-}$) and methanesulfonic acid (MSA), as

can be expected in atmospheric samples. We could also detect dimethyl sulfone ($DMSO_2$) and hydroxymethanesulfonate

(HMS) (see chromatograms in Supplement S9, Fig. S18b).

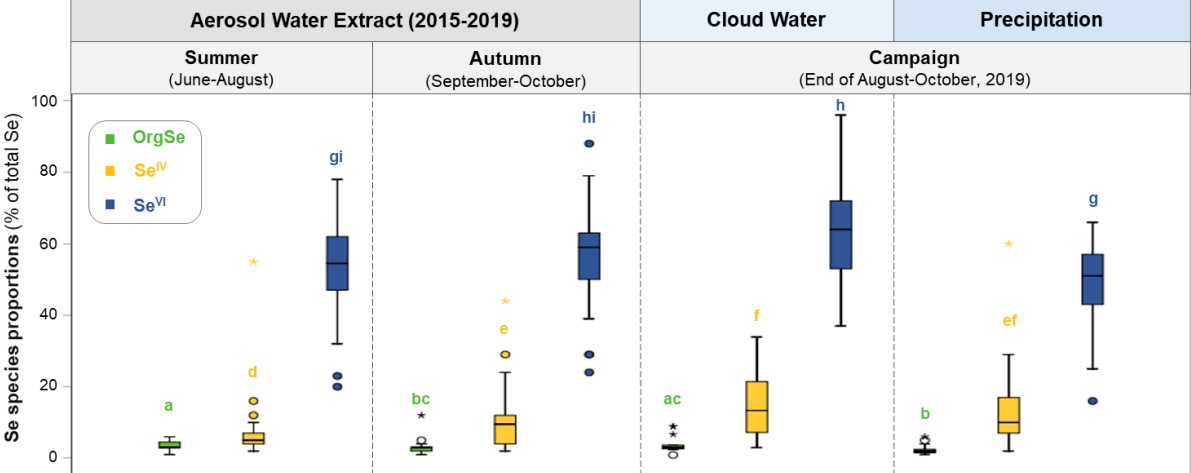

**Fig. 5.** Variability of proportions of Se species in different collected atmospheric samples. Shown Se species include: OrgSe (green), $Se^{IV}$ (yellow) and $Se^{VI}$ (blue) in the water extracts of 2015-2020 aerosol time series as well as precipitation and cloud water collected during the

campaign in 2019. Se speciation in aerosols is separately displayed for summer (June-August) and autumn (September-October). The letters (a-i) denote significant differences in species proportions (Mann-Whitney-U test; $p<0.01$).

### 3.2.2 Inorganic Se speciation is not only influenced by emission source but also by pH

To get insights into processes and/or sources affecting inorganic Se in atmospheric deposition, we investigated the species

variability in different atmospheric samples and their potential links to different source indicators. Fig. 5 shows the proportions

of Se species (% of total Se concentrations) in the water-extracts of the 2015-2020 aerosol time series as well as in precipitation

and cloud water from the campaign in 2019. No clear seasonal differences were observed for proportions of $Se^{VI}$, but $Se^{IV}$ had

significantly lower proportions in aerosol samples collected in summer (7±7%) than in autumn (11±9%).

To test for potential sources of inorganic Se, we studied correlations between inorganic speciation, moisture sources and

Py-GC-MS data in aerosols. For the 2015-2020 aerosol time series, proportions of $Se^{VI}$ positively correlate with continental

moisture sources from Spain ($r_S=0.263$, $p<0.05$), Portugal ($r_S=0.327$, $p<0.01$) and Northern Africa (crossing over the MedSea,

$r_S=0.336$, $p<0.01$). Furthermore, $Se^{VI}$ proportions positively correlate with both aliphatic and aromatic compounds ($p<0.05$),

which have been linked to various continental sources, including both urban/anthropogenic and biogenic land emissions (e.g.,

Zhao et al. (2009)). Proportions of $Se^{IV}$ showed no link to moisture sources, but positively correlate with proportions of HMS

($r_S=0.509$, $p<0.01$) and $SO_4^{2-}$ ($r_S=0.440$, $p<0.01$). HMS is formed in an aqueous-phase reaction between formaldehyde and

dissolved sulfur dioxide (e.g., Dovrou et al. (2019)). Proportions of HMS positively correlate with continental moisture sources

from Western Europe ($p<0.01$), particularly France (local source, in France south of 43.6°N). The link between $Se^{IV}$ and HMS

may indicate an anthropogenic source contributing to the $Se^{IV}$ signal in aerosols. This is also suggested from the positive





correlation of $Se^{IV}$ proportions with the abundance of (poly)aromatics ($r_S$=0.434, $p$<0.01), toluene ($r_S$=0.312, $p$<0.05) and S-compounds ($r_S$=0.353, $p$<0.01; Supplement S10, Table S10). Indeed, toluene and alkyl substituted benzenes identified by Py-

GC-MS have been previously linked to kerogen and black carbon (Zhao et al., 2009). Besides this, $Se^{IV}$ proportions negatively correlate with Py-products of proteins ($r_S$=-0.306, $p$<0.05) and levoglucosan ($r_S$=-0.354, $p$<0.01), indicating no link to fresh biomass and/or biomass burning (Fraser and Lakshmanan, 2000; Giannoni et al., 2012; Fabbri et al., 2008; Schkolnik and Rudich, 2006). To summarize, higher proportions of $Se^{VI}$ in aerosols were related to continental moisture sources and Py-compounds from land emissions, whereas for $Se^{IV}$, a likely anthropogenic contribution was observed.

Contrary to what we found for aerosols and to previous observations for precipitation samples by Suess et al. (2019), the proportions of $Se^{VI}$ in precipitation showed no link to continental sources, which is likely related to the limited number of precipitation events during our campaign with predominant moisture sources from Western Europe. Notably, the proportions of $Se^{IV}$ are larger in precipitation and cloud water samples than in aerosols (Fig. 5). Sub-events that showed relatively high proportions of $Se^{IV}$ (28-60%) in precipitation were characterized by higher moisture uptakes over coastal northern Spain (P16-

P17.1, Fig. 6a,b). Particularly coastal marine emissions seem to influence the $Se^{IV}$ signal according to the observed correlation with coastal oceanic moisture uptakes (within 30 km from the coast, only considering uptakes over the ocean; $r_S$=0.411, $p$<0.05). Notably, these relatively high $Se^{IV}$ proportions changed significantly during sub-events (see evolution of $Se^{IV}$ proportions during P17 in Fig. 6b and relative changes in moisture sources during sub-events in Fig. 6c). No HMS was detected in sub-events that showed relatively high proportions of $Se^{IV}$, indicating a different origin of $Se^{IV}$ in wet deposition than for

the 2015-2020 aerosol dataset.

The long-term weekly resolved aerosol and high-resolution precipitation data thus indicate different sources for $Se^{IV}$. The 2015-2020 aerosol time series mainly has an anthropogenic signature of $Se^{IV}$, whereas the (sub)event-based sampling in 2019 enabled elucidating other sources and processes. In particular, marine coastal emissions (potentially in relation to sea spray emission) seem to contribute to higher $Se^{IV}$ proportions in wet deposition.





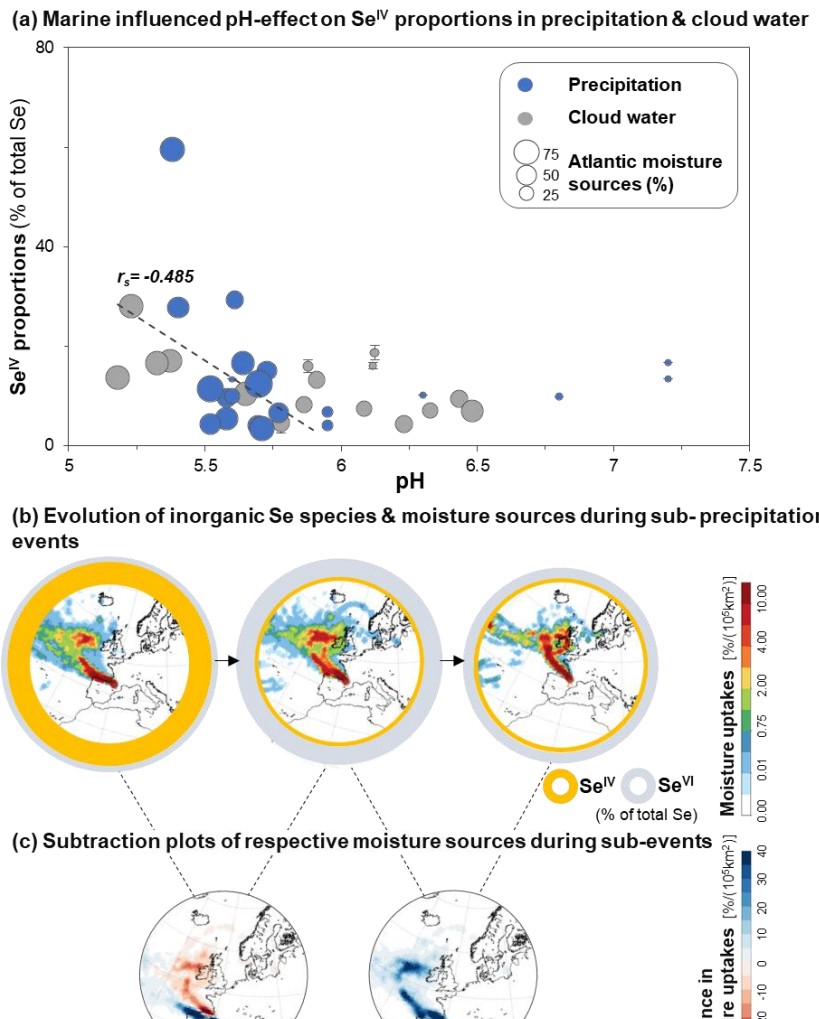

**Fig. 6.** Inorganic Se speciation in precipitation and cloud water as a function of dominating moisture sources. **(a)** Proportions of Se[IV] (% total Se) as a function of pH in precipitation (blue circles) and cloud water (grey circles) during the high-resolution campaign in 2019. Size of data points correspond to proportions of Atlantic moisture sources (% total moisture sources). **(b)** Evolution of proportions of inorganic Se species, Se[IV] (yellow) and Se[VI] (grey) during sub-precipitation events (P17.1: 270 min, P17.2: 380 min, P17.3: 110 min duration) with corresponding moisture source plots (shown in colour scale 0-10 %/$10^5$ km$^2$). **(c)** Subtraction plots of relative moisture source changes (shown in red-blue colour scale from -40 to 40 %/$10^5$ km$^2$) during sub-precipitation events shown in (b).

Although we can link inorganic Se speciation to specific marine/coastal sources, we further found an effect of pH in wet deposition (pH in aerosols was not determined). Oxidation-reduction reactions are strongly influenced by the redox potential (pe) and pH in natural waters. From the pe–pH stability field diagram (Séby et al., 2001), it is clear that the speciation of Se is likely quite sensitive to pH under atmospheric conditions. For a redox potential of pe~10 (range of rainwater pe: 8-11; Willey et al. (2012)), Se[IV] is the thermodynamically favoured species at acidic pH values (~2.5-5.5), whereas Se[VI] is favoured for pH



values >6 (Séby et al., 2001). Fig. 6 shows the proportions of $Se^{IV}$ in relation to pH of precipitation (blue circles) and cloud water (grey circles). Precipitation and cloud water with predominant Atlantic moisture sources (size of data points correspond to percentages of Atlantic moisture sources) show a negative correlation between pH and $Se^{IV}$ proportions ($r_S$=-0.485, $p$<0.05),

indicating a marine influenced pH-effect on Se speciation in precipitation and cloud water. Although sea salt aerosols are generally thought to be in the alkaline pH range (Pye et al., 2020), a recent study showed that pH values of freshly emitted sea spray aerosols are significantly lower than that of sea water (approximately pH 4 versus pH 8), with even lower pH values for aerosol particles below 1 µm in diameter (Angle et al., 2021). Thus, the observed link between pH and $Se^{IV}$ proportions in precipitation events with predominant Atlantic moisture (Fig. 6a), might be related to the influence of sea spray. To further

investigate potential influences of atmospheric sources on inorganic Se speciation, we did a literature review of previously reported ratios of $Se^{IV}$ to $Se^{VI}$ and classified the sampling sites into urban, industrial, marine, forested area and remote (Fig. 7). Higher proportions of $Se^{IV}$ have been reported in urban, industrial and marine environments ($Se^{IV}$: $Se^{VI}$= 2.2±2.4) compared to the precipitation samples collected at Pic du Midi (this study; $Se^{IV}$: $Se^{VI}$=0.4±0.7) as well as ratios reported in remote and forested areas ($Se^{IV}$: $Se^{VI}$=0.4±0.3). Despite the small number of samples in referred study, the relatively high ratios previously

reported in a marine environment (western North Atlantic) (Cutter and Church, 1986) agree with our observations above, i.e., higher proportions of $Se^{IV}$ for precipitation events with predominantly Atlantic moisture.

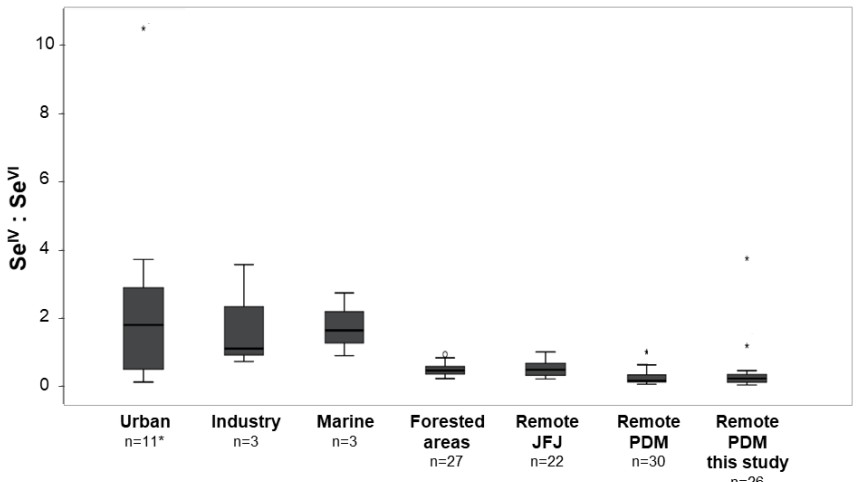

**Fig. 7**. Previously reported ratios of $Se^{IV}$ to $Se^{VI}$ concentrations classified by sampling sites: urban, industry, marine, forested area, remote. Number of data points indicated for each source class is indicated by n. *Urban data derived from 5 different studies (Cutter, 1978; Cutter
and Church, 1986; Wang et al., 1994; Wallschläger and London, 2004; Donner and Siddique, 2018). Other classifications comprise data from industry (De Gregori et al., 2002), marine (Cutter and Church, 1986) and forested areas (Roulier et al., 2021). Remote data refers to this study and previous study by Suess et al. (2019) conducted on two high altitude stations: Jungfraujoch (JFJ, Switzerland) and Pic du Midi (PDM, France).

### 3.2.3 Sources of organic Se

We further investigated potential source signatures of the OrgSe fraction in the 2015-2020 aerosol time series. Proportions of OrgSe in aerosols collected in summer (3.5±1.3%) had higher proportions than aerosols collected in autumn (2.8±2.0; Fig.




5). Similar to OrgSe, higher MSA concentrations were observed in summer compared to autumn (i.e., 5.3±2.5 versus 2.3±1.3 ng·m$^{-3}$), and the proportions of MSA and OrgSe ($r_S$=0.369, $p$<0.01) correlate significantly, suggesting that OrgSe may originate from a similar biogenic source as MSA, which is an oxidation product of DMS (Chen et al., 2018).

To further investigate a potential biogenic source of OrgSe in aerosols, we studied links to dominant moisture sources and other identified organic compounds by Py-GC-MS. Proportions of OrgSe in the water-soluble aerosol extracts correlate with Atlantic moisture sources ($r_S$=0.242, $p$<0.05), similarly to MSA ($r_S$=0.328, $p$<0.01). Furthermore, the proportions of OrgSe positively correlate ($r_S$=0.532, $p$<0.01; Supplement S10, Table S10) with the relative abundance of Py-products that are specifically derived from proteins and amino acids (i.e., 2,5-diketopiperazine noted DKP (Voorhees et al., 1997; Voorhees et

al., 1992; Fabbri et al., 2012).

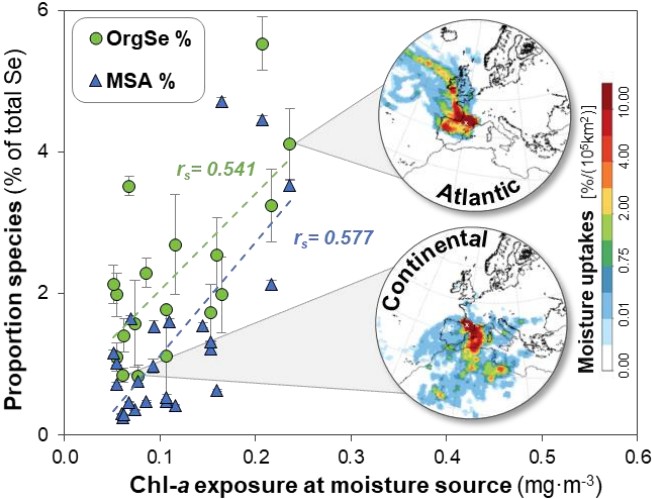

**Fig. 8.** Proportions of OrgSe (green circles) and MSA (blue triangles) as a function of Chl-*a* exposure at moisture source of precipitation. Two examples of low and high values of OrgSe proportions are shown with corresponding moisture plots. Example of high OrgSe proportions shows high Atlantic moisture sources (37% of total moisture sources, P5.2), whereas low OrgSe example is dominated by
continental moisture (22% North Africa and 44% Western Europe, P9.2). It should be noted that moisture uptakes are displayed as % uptakes per area (10$^5$km$^2$), thus depending on defined size of areas, high moisture uptakes (red) don't necessarily correspond to predominant sources. The error bars represent the standard deviation values resulting from quantification by LC-ICP-MS/MS in duplicate.

For the precipitation and cloud water samples collected during the campaign in 2019, there is no significant correlation between the proportions of OrgSe and Atlantic moisture sources, in contrast to MSA ($r_S$=0.705, $p$<0.01) and DMSO$_2$
($r_S$=0.730, $p$<0.01). Instead, the proportions of OrgSe and chlorophyll *a* (Chl-*a*) exposure at the moisture source ($r_S$=0.541, $p$<0.05; Fig. 8) significantly correlate. Both MSA ($r_S$=0.577, $p$<0.01; Fig. 8) and DMSO$_2$ ($r_S$=0.494, $p$<0.05) also show significant correlations to Chl-*a*, which was expected, since they originate from a known (marine) biogenic S source (Harvey and Lang, 1986; Pszenny et al., 1990; Watts et al., 1987). Further analysis of the Chl-*a* content showed that exposure originated almost completely from the Atlantic (particularly Bay of Biscay), with little contributions from the subtropical North Atlantic
and the Mediterranean Sea (approximately 6% of total exposure) and no detectable exposure over land from lakes and wetlands.



Both results from the 2015-2020 aerosol time series and the 2019 campaign generally imply that OrgSe originates from a marine biogenic source. The identified OrgSe may represent oxidation products of DMSe and/or other methylated selenides. OrgSe in aerosols may also originate from a marine particulate source such as bacteria and algae cells or from associated biomolecules emitted to the atmosphere together with seaspray (similar to the Py-products of proteins) (Aller et al., 2017).

**4    Conclusions and environmental implications**

Our approach of combining multiple measurement and modelling techniques to study the atmospheric trace element cycling offers new insights into the complex processes influencing the atmospheric deposition of trace elements in general, and Se in particular. Past research in this field has often been based on the assumption that significant positive correlations between elemental concentrations in atmospheric deposition solely reflect specific source contributions. Our data shows that

total Se concentrations (and other elements) in atmospheric deposition rather mirrors a combination of contributing source emissions, atmospheric processing during local versus long-range transport and local cloud dynamics associated with specific weather events (i.e., deep convective activity during thunderstorms or large-scale uplift during frontal passages). We demonstrate that aerosol loading and elemental concentrations in wet deposition were highest during thunderstorms at our sampling site, thereby impacting the supply of micronutrients and potentially toxic elements to surface environments.

We show that speciation techniques in combination with other chemical proxies, and trajectory modelling offer a unique tool for separating source contributions and atmospheric processes affecting Se concentrations. The main Se species, $Se^{VI}$ is primarily linked to continental sources and for the first time we could identify an organic Se species as a biomarker for marine-biogenic sources. Identified $Se^{IV}$ in aerosols had a primarily anthropogenic source signature, whereas coastal emissions seem to particularly contribute to $Se^{IV}$ proportions in precipitation. Nevertheless, it should be noted that inorganic Se species ($Se^{IV}$

or $Se^{VI}$) are also affected by atmospheric processing, such as atmospheric oxidative transformations and pH changes, thereby losing their source information with increasing distance to emission sources. Therefore, it is likely that $Se^{IV}$ indicates more local sources, whereas $Se^{VI}$ could also be transported over longer distances.

Since $Se^{IV}$ and $Se^{VI}$ are the dominant species in atmospheric deposition, they are of particular interest for assessing the fate of deposited Se in soil-plant systems. Selenium uptake by plants is largely controlled by its concentration, its speciation

and the concentration of other chemical species competing for plant-uptake (Winkel et al., 2015). On a global level, atmospheric inputs largely exceeding Se inputs from bedrock according to a previous estimate (870 vs. 35 $mg \cdot ha^{-1} yr^{-1}$; Feinberg et al. (2020a)), although regionally, Se can be predominantly of geogenic origin or derived from fertilizers. Due to chemical similarity, the more mobile species, i.e., $Se^{VI}$, is taken up via the $SO_4^{2-}$ transporter by plants and thus competes with $SO_4^{2-}$ for plant uptake (Winkel et al., 2015). We investigated the ratio of $Se^{VI}$ to $SO_4^{2-}$ in precipitation from different moisture sources

and found that predominantly Atlantic moisture sources led to the highest ratios of plant-available Se (Supplement S10, Fig. S21). Therefore, even though Atlantic moisture sources generally supplied lower Se levels than continental moisture sources for our study site, it supplied the highest fraction of bioavailable Se in precipitation. This understanding is important in the





context of predicted changes in emission sources (e.g., shifts in anthropogenic versus natural emissions, Feinberg et al. (2021)) and deposition of Se as micronutrient to surface environments. Furthermore, the finding of coupling versus uncoupling

of hydrological and trace element cycling based on stable water isotope analysis represents an interesting research avenue considering expected future changes in precipitation patterns and how these will affect atmospheric deposition of (trace) elements.



**Data availability**

All chemical data of this study are available on the ETH Research Collection with the DOI xxx.

**Supplement**

The supplement related to this article is available online at xxx.

**Authors contributions**

E.S.B., J.T and L.H.E.W. conceptualized the study, with inputs from H.W., F.A., I.T., and A.F. E.S.B. created visualizations, and wrote the manuscript with contributions from all the co-authors. J.E.S. conceptualized the observational setup and provided all aerosols samples. V.P. managed all observational measurements of particle number, black carbon and meteorological data onsite. E.S.B. and I.T. performed the fieldwork during the 2019 campaign. E.S.B. carried out sample preparation, chemical analysis and data treatment with inputs from J.T. and S.B. J.T. conducted the Py-GC-MS analysis and data treatment. F.A. provided air parcel back-trajectories, moisture sources and Chl-*a* data and A.F. performed all SOCOL-AER simulations. E.S.B. conducted all statistical analysis with inputs from J.T. E.S.B. interpreted all findings with help from J.T., L.H.E.W. and with contributions from I.T., F.A., A.F., S.B., J.E.S, V.P. and H.W.

**Competing interests**

At least one of the (co-)authors is a member of the editorial board of Atmospheric Chemistry and Physics.

**Acknowledgments**

Samples and observation data were collected at the Pyrenean Platform for Observation of the Atmosphere P2OA (https://p2oa.aeris-data.fr). We especially acknowledge Francois Gheusi and and Serge Soula for managing observational measurements onsite and maintaining the P2OA database, as well as the technical support from the UMS 831 Pic du Midi Observatory team. The authors are grateful to their colleagues at Eawag and ETH, especially Björn Studer for carrying out the stable water isotope analysis, Elyssa Beyrouti for her contributions to the Py-GC-MS measurements, Pauline Béziat for providing the DMSeP standard and Caroline Stengel for general lab support.

**Financial support**

This study was funded by the Swiss National Science Foundation (project number 179104) and internal funds from Eawag and ETH Zurich. I.T. was partially funded by the Swiss Polar Foundation (project DAWATEC). P2OA facilities and staff are



710  funded and supported by the University Paul Sabatier, Toulouse, France, and CNRS (Centre National de la Recherche Scientifique). P2OA is part of the national research infrastructure ACTRIS-France.





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
