# Peer review of "Influences of sources and weather dynamics on atmospheric deposition of Se species and other trace elements"

_EGUsphere, 2023_

## Referee Comment (RC2)

Referee comments on the article: egusphere-2023-1135:

Influences of sources and weather dynamics on atmospheric deposition of Se species and other trace elements

By Esther S. Breuninger, Julie Tolu, Iris Thurnherr, Franziska Aemisegger, Aryeh Feinberg, Sylvain Bouchet, Jeroen E. Sonke, Véronique Pont, Heini Wernli, and Lenny H. E. Winkel

General comment

In the paper, the authors present a comprehensive study of the main factors describing the atmospheric deposition of Se. For this purpose, the authors present experimental measurements that are correlated with model results. It is important to highlight the large number of samples collected for the analysis (weekly samples during 5 years), as well as the precipitation samples obtained. Another interesting aspect of this work is the analysis performed at the high-altitude atmospheric observatory (Pic du Midi (French Pyrenees; 2877 m a.s.l.), which allows the analysis of the synoptic transport of Se. I consider that this article can be accepted in its present form because it is well-written and presents very interesting results.

Here are some minor details and suggestions.

Specific comments

- I recommend the authors write a shorter abstract. Please, introduce a more summarized writing where you highlight the main findings of the article. I think this paper presents a very complete analysis, but if the authors abbreviate this section, it makes it more appealing to the reader.
-
- Line 37 The authors state that trace elements and isotopes in water are decoupled from clouds during precipitation. Can the authors indicate the specific result where this effect is observed?

- This is a very long article, as it presents different subsections. I recommend introducing a table of contents at the beginning of the manuscript.

- Methods section: the reviewer strongly recommends the authors extend the description of the study region, specifically to detail the Pic du Midi Observatory monitoring station site. It would be interesting to know the height of the mixing layer in the region. Since this study deals with a synoptic transport of Se, it would be interesting to know if the authors discarded the local influence (convective transport in the mixed layer).
- From your results: ¿it is possible to assess if Se concentrations decline with precipitation amount?
- What about the solubility of Se species: your study, only measures dilute Se species?
- In the conclusions, you state that: "for the first time we were able to identify an organic Se species as a biomarker of marine biogenic sources". Could you describe the specific result that allows you to conclude this?
- Line 365: Se can be released to the atmosphere from a variety of sources (e.g., natural sources such as soil, water, and vegetation, as well as anthropogenic sources). Are the terrestrial contributions of Se at Pic du Midi only due to synoptic transport or to local sources?

---

## Author Comment (AC1)

**Author response to all referees' comments on "Influences of sources and weather dynamics on atmospheric deposition of Se species and other trace elements"**

We would like to thank the three reviewers for their time reviewing our manuscript and their valuable comments, which significantly enhanced the clarity of the paper. We have considered all comments carefully and present our point-by-point responses below (reviewer comments are in blue and author responses are in black). In the manuscript and supplementary information, the changes are shown using the track changes feature of word processor.

**Response to Referee #1**

**Sample processing (lines 172-173, lines 182-184, line 192) and Chemical analysis (lines 215-218):** from the methods description given, I understand that the conditions of sample treatments (extraction of water-soluble fraction of aerosol samples and pre-concentration of these fractions and of precipitation/cloud water samples) was the same before speciation analyses of Se and S species. However, details of method development given in Supplement S2 (in particular species stability during extraction and pre-concentration) only target on Se species and no information about the stability of S species is provided. This may be due to the fact that such extraction and/or pre-concentration methods were previously validated for S species? In that case corresponding reference(s) should be added. Otherwise, the whole discussion around identified S species may be somewhat more complicated.

Thank you for pointing out that the pre-treatment of samples before S speciation analysis was unclear. S species were measured in precipitation samples without pre-concentration because their concentrations are high enough. The optimized pre-concentration method was developed to enable the quantification of Se species only. We have now clarified this in the method section on P10, L222-227 as follows:

"*Sub-samples used for Se speciation analysis were pre-concentrated by lyophilisation of frozen samples from an initial volume of 12 mL (for precipitation) or 9 mL (for water extract of aerosol filter) to a residual volume of 1.5 mL (pre-concentration factor of 8 or 6) to which ammonium citrate was added to increase ionic strength. The initial volume of cloud water samples was variable depending on sampled amounts. Further details on optimized protocol and stability of Se species during pre-concentration are given in Supplement S2. S speciation was analysed directly in all atmospheric samples (i.e., without the pre-concentration step).*"

**Lines 236-245:** the quantification method used for Se and S species must be specified in this section.

We agree with the reviewers. We have now clarified the quantification method used for Se and S species on P11, L252-255 as follows:

"*Quantification of Se and S species was done by external species-specific calibration (i.e., mixed Se or S species standards prepared in the corresponding LC eluent). Data treatment was performed using the MassHunter 4.6 (Agilent) software. SeCys$_2$ was used for the quantification of the "OrgSe" peak by anion exchange as further described in section 3.2.1.*"

**Line 351:** as the indication of the range of values is given in previous sentence, the median value would be more meaningful here.

We agree with the reviewer and included both the average and median values on P15, L362-366:

"*Total Se concentrations (after digestion using HNO$_3$ and H$_2$O$_2$) ranged from 2 to 221 pg·m$^{-3}$ in the 2015-2020 aerosol time series (n=134; average: 58±57 pg·m$^{-3}$, median: 33 pg·m$^{-3}$; Fig. 2 and in Supplement S5, Fig. S4). We clearly see higher Se concentrations in summer (June-August; average:*"

*114±50 pg·m$^{-3}$, median: 111 pg·m$^{-3}$) than during spring (March-May; average: 32±24 pg·m$^{-3}$, median: 30 pg·m$^{-3}$), autumn (September-November; average: 49±44 pg·m$^{-3}$, median: 27 pg·m$^{-3}$) and winter (December-February; average: 9±5 pg·m$^{-3}$, median: 9 pg·m$^{-3}$)".*

We thought it was important to keep the average value to enable (future) comparison with the literature since most previous studies solely reported average concentrations.

**Lines 519-533 and corresponding supplements: Lines 519-533 and corresponding supplements**: the comparison of example Se chromatograms illustrating the presence of a third Se-containing peak (fig S18a) to the standards one (supplement S3) seems to indicate a shift in retention times, although retention time units being different. It is not obvious to distinguish if the retention time of Org Se is different from the one of selenocystine standard? To facilitate readability, it would be helpful if chromatograms are plotted with same units (select min or s, and counts or counts per second for all layouts).

Thank you for pointing out this inconsistency in showing chromatograms. As proposed, the unit for retention time now made consistent in all chromatograms to seconds (sec). Please, see Figs. S3, S18 and S19 in the revised Supplement. The retention time of OrgSe is not different to the one of SeCys$_2$. For clarification, we added the following text in the caption of Fig. S18 (S9, P25):

*"Quantification of Se and S species was done by species-specific external calibration with mixed Se or S species prepared in the corresponding LC eluent. SeCys$_2$ was used for the quantification of OrgSe peak given that both had the same retention time (i.e., 50s)."*

The complementary use of cation exchange chromatography to try to understand the nature of Org Se peak is interesting. However, from Fig S19, the result of DMSeP addition is not quite clear as 2 peaks are increasing and one is decreasing. Then Se species recoveries are given (lines 524-525) but as indicated in previous comment, species quantification method was not indicated, which is important to know in particular in the case of Org Se peak, for which no commercial standard is available, to clarify recoveries and concentrations values (lines 530-533).

For the complementary analysis of DMSeP, we used an in-house standard synthesized by the protocol of W.-M. Fan et al. (1998) (DMSeP validated by Orbitrap and NMR analysis). The synthesis has several by-products of other organic Se compounds, which are visible in the spiked sample in Fig. S19, thus making the quantification of DMSeP difficult.

This is why we had detailed in the main manuscript that i) one peak "*co-elutes with an in-house synthesized standard of the metabolite dimethylselenonium propionate (DMSeP), which could indicate that this peak consists of small methylated Se compound(s)*" (P22, L539-541), and that ii) "*To achieve full identification of the organic Se species, high-resolution mass spectrometry may further be used given that sufficient pre-concentration can be reached*" (P22, L544-546).

Regarding the concentrations and recoveries for OrgSe, as described in the answer to the comment on L236-245, SeCys$_2$ was used for the quantification of the OrgSe peak by anion exchange chromatography because OrgSe had the same retention time than SeCys$_2$. The recoveries the reviewer is referring to were then calculated by the sum of identified species (i.e., OrgSe, Se$^{IV}$, and Se$^{VI}$) via anion exchange chromatography coupled to ICP-MS compared to total Se in the samples (i.e., precipitation, cloud water or aerosol water extracts).

**Lines 539-541 and corresponding supplement:** examples of S chromatograms of fig S18b indicate poor chromatographic separation of HMS (hydroxymethanesulfonate) and MSA (methanesulfonic acid)

peaks for which resolution does not allow accurate compounds quantification thus calling into question the following discussion and correlations using proportions or concentrations of these S compounds.

Indeed, these two S species are not perfectly separated. The LC-ICP-MS/MS method we used, which was developed by Müller et al. (2019), is the only available method to analyse S species at the low S concentrations encountered in atmospheric samples and further method optimizations did not help to improve the separation. To ensure correct quantification, all calibration standards were prepared with species-specific standard mixtures (including MSA and HMS). The Masshunter software allows consistent quantification, if the same peak integration conditions are kept for calibration standards and samples.

Furthermore, the quantified HMS and MSA species showed different links to other analysed and independent datasets, which is consistent with previous studies, i.e., HMS correlated with continental moisture sources, consistent with its expected anthropogenic origin, and MSA correlated with marine moisture sources, consistent with its expected marine origin.

As mentioned in our answer to a comment above, we have clarified the quantification method for S species on P11, L252-255: "*Quantification of Se and S species was done by external species-specific calibration (i.e., mixed Se or S species standards prepared in the corresponding LC eluent). Data treatment was performed using the MassHunter 4.6 (Agilent) software.*"

Furthermore, we added the following clarification in the discussion on P23, L558-560:

*"To ensure correct quantification of MSA and HMS, which elute very close to each other, species-specific calibration standards (i.e., mixed S species standards) were used."*

**Lines 551-552**: Se$^{IV}$ proportion in aerosol water extracts appears to be significantly lower in summer than in autumn samples, does this difference remain if proportion is calculated as % of total Se in aerosol instead of its water-soluble extract?

Similar to the species proportions discussed on P23, L571-572 (calculated as % total Se in water soluble extract), the Se$^{IV}$ proportions are significantly lower in summer (5±4%) than in autumn (9±7%), when the proportions in the aerosol water extracts are calculated as % of total Se in the total aerosol (i.e., in the acid digest). We specifically optimized the aerosol water extraction of Se for an extraction volume (15 mL) that is sufficient for both speciation analysis (after pre-concentration) and total analysis. If concentrations are compared between different extracts (i.e., water extract vs aerosol digest), there is a risk of introducing a certain bias that is caused by the general variability of Se over the entire aerosol filter.

**Lines 558-562**: as indicated in previous comments, without information about S species stability during extraction (and if so, preconcentration) and given the poor resolution between HMS and MSA peaks, the discussion involving HMS is very hypothetical.

See our answer to comment on Lines 539-541 & about method. Briefly, S speciation was analysed directly, i.e., without the pre-concentration step.

**Lines 571-572**: precipitation/cloud waters were collected in the period from end of August to October, comparison with aerosol water extracts of the same period (September to October) only indicates a difference between aerosol extracts and cloud waters, while Se$^{IV}$ proportion in precipitation waters was not significantly different neither from cloud waters nor aerosol extracts (from letters indicated in fig 5).

We agree with the reviewer and have thus removed this sentence to avoid confusion.

**Lines 622-624**: same comment for discussion around MSA as previously indicated for discussion involving HMS.

See our answer to comment on Lines 539-541 & about method.

**Line 662**: Org Se was detected and not identified, the presence of single species in this chromatographic peak eluting at or close to the void volume of the column was not proven here

We agree with the reviewer. Therefore, we modified the sentence "*for the first time we could identify an organic Se species as a biomarker for marine biogenic sources*" as follows: "*for the first time we could detected a new Se fraction likely of organic nature, which appears to be a biomarker for marine biogenic sources. Further work is required to investigate the molecular composition of this Se fraction and its role in atmospheric Se cycling.*" on P28, L683-685.

**Supplement S2**:

**Line 59:** partial transformation of SeMet is indicated to explain its lower recovery following extraction of water-soluble fraction of aerosols. It would be interesting to learn more (chromatographic detection of other Se-containing compound(s) and if so, which retention time(s)? ).

Yes, the initial spiked SeMet was transformed into a Se species that eluted close to the void volume of the anion exchange column and so close to the retention time of $SeCys_2$ (retention time: 50s). Since we did not identify SeMet in analysed atmospheric deposition samples, we did not investigate specific transformation products of SeMet further. These transformation products are assumable very specific to SeMet (e.g. selenomethionine-oxide, SeOMet) and thus should not have an impact on our results. Nevertheless, we agree that it is a very interesting question that could be further investigated by high-resolution mass spectrometry given that sufficient pre-concentration can be reached.

**Lines 87-88:** lyophilisation to dryness led to losses of organic Se species for which the authors indicated a possible "transformation to other organic Se species that are not retained by anion exchange". This is a very important point as compounds not retained by anion exchange should elute at or close to the void volume of the column, as it is the case for the unknown organic Se compound/pool later detected in samples.

This is correct. However, this observed transformation was very specific to the lyophilisation to dryness in low ionic strength matrices (ultrapure water and rainwater) and was not observed in the optimized matrix (ammonium citrate) later applied to collected samples as described in Supplement S2, P5-7. For clarification, we have now added a sub-section giving the optimal conditions applied to the samples (see section 2.3).

**Lines 97-104:** the second set of tests compared different containers using lyophilisation to a residual volume < 1,5 mL. In this part, it is not clear in which aqueous medium (or media) the test was done? In the same way, tested media (lines 74-78) were ultrapure water, ammonium citrate solution and rainwater, but water-soluble fraction of aerosols does not seem to have been considered, is that right?

These specific tests were done in ammonium citrate solution as indicated in the caption of Fig. S2 ("*Lyophilisation was done with addition of 2 mmol L$^{-1}$ ammonium citrate solution (eluent of LC-method*"). For clarification, we have now added this information in the text, S2, P6, L102-103 as follows:

"*Compared to when using lyophilisation of sample to complete dryness, all Se species in 2 mmol L$^{-1}$ ammonium citrate solution were entirely recovered with lyophilisation to a residual volume of <1.5 mL, and this was true for all tested containers*".

The stability of different Se species during the extraction from aerosol filter samples are described in Supplement S2.1. For the following lyophilisation experiments, the water-soluble fraction of aerosols was not considered since rainwater samples and aerosol water extracts are expected to have very similar matrices. Furthermore, the performed lyophilisation tests showed that ionic strength was the major cause of species transformation. Since all collected aerosol samples were pre-concentrated with the addition of ammonium citrate, no significant differences in stability are expected between tested rainwater samples and the water-soluble aerosol fraction.

**Supplement S10:** add in the table caption, the number of samples considered for calculated correlations. It is important information as although statistically significant with p-values < 0.01 or <0.05, the strength of correlations is weak to moderate.

We have added the number of samples considered for the calculated correlations in the caption of Table S10.

**Technical corrections**

- Abstract line 19: check for duration of aerosol sampling 2015-2019 or 2015-2020? done
- Line 167: "53.6 ± 2.8" round to the number of significant figures. done
- Line 205: revise "equipped with and SPS4…" done
- Line 321: revise "in order to minimized…" done
- Lines 352-354, 530-532 and 536-538: round to the number of significant figures done
- Fig 5: aerosol time series is 2015-2019 in the figure, 2015-2020 in figure caption done
- Supplement S3 line 127: revise "an anion exchanges…" done
- Table S3: round to the number of significant figures (recoveries column) done
* * *
**References**

Müller, E., von Gunten, U., Bouchet, S., Droz, B., and Winkel, L. H. E.: Hypobromous Acid as an Unaccounted Sink for Marine Dimethyl Sulfide?, Environmental Science & Technology, 53, 13146-13157, 10.1021/acs.est.9b04310, 2019.

W.-M. Fan, T., N. Lane, A., Martens, D., and M. Higashi, R.: Synthesis and structure characterization of selenium metabolites†, Analyst, 123, 875-884, 10.1039/A707597I, 1998.

---

## Author Comment (AC2)

**Author response to all referees' comments on "Influences of sources and weather dynamics on atmospheric deposition of Se species and other trace elements"**

We would like to thank the three reviewers for their time reviewing our manuscript and their valuable comments, which significantly enhanced the clarity of the paper. We have considered all comments carefully and present our point-by-point responses below (reviewer comments are in blue and author responses are in black). In the manuscript and supplementary information, the changes are shown using the track changes feature of word processor.

**Response to Nadine Borduas-Dedekind & team**

This paper represents a large undertaking by the team of authors of method development, monitoring data, reporting of unique Se concentration datasets, hypothesis-generating analysis related to the role of thunderstorm clouds AND climate modeling. It's a super paper. The reported data is of the highest quality, the methods are accurate, and the details included in this paper are remarkable. This manuscript also includes a highly detailed SI.

Creation of their aerosol sample (n=134) at weekly resolution over 5 years is impressive.

The figures in this paper are remarkable! Some of the best made figures we've seen. The TOC figure is clean, clear, precise, and really compelling. Figure 1 is so clean and clear and yet has a ton of information portrayed. The use of symbols is particularly clever. Figure 6b also allows an unusually large amount of information to be presented clearly.

The results presented in Figure 4 are particularly compelling. The enhancement due to thunderstorms is clear (and unexpected to us).

(section 3.2.1) The measurement of the "third peak" as organo-selenium compound was particularly exciting for our group (since we are working on atmospheric methylated selenium compounds and their atmospheric fate). And the authors present convincing evidence of the presence of reduced Se species.

We would like to thank Nadine Borduas-Dedekind and her team for taking the time to read our article in detail and for providing constructive feedback. Thank you for acknowledging the broad and high-quality datasets we acquired as well as appreciating the quality of the figures.

We felt there were two stories in this paper: Se concentrations over 5 years (Fig 2), and trace elements in atmospheric deposition/role of thunderstorms (Fig 3). It makes for a super paper; but it wasn't clear to us, the readers, why these stories needed to be present in one super paper. If the authors could connect the two storylines better, it would be beneficial for the future reader.

In this study, our overarching goal was to bring significant contribution to an improved understanding of atmospheric Se deposition as stated in the introduction, P6, L124-125.

"*Our objective in this study is to comprehensively investigate the role of different source factors and weather dynamics on the observed variability in trace element concentrations and Se speciation in atmospheric deposition.*"

We used different datasets (weekly aerosol series, high-resolution campaign of precipitation and cloud water samples) to address this overarching objective. The high-resolution campaign offered the investigation of unique source and process factors within individual precipitation events, whereas the aerosol series offered more long-term insights, however at lower temporal resolution. The effects of local cloud dynamics of specific weather events on total element deposition as well as the effect of source and process factors on Se speciation in atmospheric deposition samples are two linked research gaps, which significantly contribute to an improved understanding of atmospheric Se deposition. Splitting the material in two papers would, in our view, weaken the paper(s) substantially. Furthermore,

it was important to not only analyse Se and its specific source signatures (Fig. 2), but also other (trace) elements (Fig. 3) to highlight process factors (i.e., as identified for deep convective activity during thunderstorms) that affect multiple elements.

For clarification, we added the following sentence on P18, L448-451:

 *"Even though our study focuses on Se, we also investigated other (trace) elements in wet deposition which have been used as source indicators in previous studies, e.g., mineral dust (e.g., Fe and Mn) or anthropogenic activities (e.g., Pb)."*

References to further support lines 60-62 discussing oxidation of volatile selenium should include (Atkinson et al., 1990) and could also include our very recently accepted work: https://doi.org/10.1021/acs.est.3c01586 (in press, Heine & Borduas-D., ES&T, 2023). We observed selenic acid and dimethyl selenoxide as products. Same references are also relevant for discussion in lines 646-649.

Thank you for the suggestion to include these references on oxidation of volatile selenium. We included these references on P4, L64-65 and on P28, L667-668.

The discussion about variability of Se over the 5 years dataset in section 3.1.1. is likely location-dependent. We saw in (Lao et al., 2023) that the seasonal trends were drastically different and could be categorized in 6 distinct profiles related to geographical location in the US. Did the authors consider normalized their Se data with PM total mass? That analysis would help identify times of Se depletion or enhancement events (Lao et al., 2023).

We wanted to do so, but the aerosol mass on our collected filters could not be determined gravimetrically due to low aerosol loading and relatively large filter size (i.e., aerosol mass within the balance error of the filter mass). The aerosol loading under free tropospheric conditions is very low in contrast to most sampling sites of monitoring networks, which are generally located at low altitude sites (as is the case for the IMPROVE network in Lao et al. (2023)). No online measurements of the particle mass were available at the monitoring station.

We were therefore curious about the "similar patterns" referred to on lines 356-358. Could the authors show these similarities in their SI for example?

The similar patterns you referred to are already illustrated in Supplement 5, Fig. S5 (P15). For this figure, we selected three elements (S, Fe, and Pb) because of their likely contrasting source origins. Reference to Fig. S5 was given in the sentence following the one you referred to:

*"This is interesting as it is likely that these elements are derived from different sources, e.g., mineral dust (e.g., Fe and Mn) or anthropogenic activities (e.g., Pb, Supplement S5, Fig. S5)"*.

In order that future readers do not miss Fig. S5, we modified the text on P15, L369-373 as follows:

*"In our study, we compared Se concentrations to many other (trace) elements and most of them show a similar pattern with higher concentrations in summer (see data for S, Fe and Pb in Fig. S5). This is interesting as it is likely that these elements are derived from different sources, e.g., mineral dust (e.g., Fe and Mn) or anthropogenic activities (e.g., Pb), which would indicate that elemental concentration patterns are not only driven by sources but also other (physical) processes"*.

**Lines 400-405.** We thought it would be worth tabulating this information from the literature. It would be a great resource to point to and build upon for the authors, but also for the community.

The papers of Feinberg et al. (2020b); (2020a), which we cited on P4, L66-72, already tabulated Se concentrations in precipitation and aerosols from previous studies in their supplementary information. We did not want to duplicate this effort. We have however gathered and presented published Se speciation in precipitation in Fig. 7 as this has not been done in a previous study.

**(section 3.2.2) lines 551-552 and subsequent paragraph**: do the authors have an idea why there was no clear seasonal difference for Se(VI)? We thought that correlations with aromatics and aliphatics which likely oxidize quickly was surprising. What role of aerosol partitioning, aerosol viscosity could play here? Se can also undergo fast redox, and could Se(IV) and Se(VI) undergo redox chemistry in the atmosphere? See for example: (Reich and Hondal, 2016)

We think that there is no clear seasonal difference for $Se^{VI}$ because $Se^{VI}$ is the oxidation product of various emitted Se species. Regarding the correlation of $Se^{VI}$ with aromatics and aliphatics, it is important to keep in mind that the aromatic and aliphatic compounds detected by Py-GC-MS are not necessarily aromatic and aliphatic compounds present in the aerosols, but rather pyrolytic products of specific compounds. Indeed, Py-GC-MS allows identifying pyrolytic products of organic matter, which are specific for biochemical classes of organic matter, such as carbohydrates, proteins (detailed list in Supplement, S4), and/or have been previously related to certain compounds, sources or conditions. For example, (poly)aromatic and aliphatic compounds detected by Py-GC-MS have been related to anthropogenic sources (Zhao et al., 2009; Subbalakshmi et al., 2000) and organic compounds that do not oxidize quickly (e.g., lipids, black carbon (Zhao et al., 2009; Zhao et al., 2012). Therefore, the correlations of $Se^{VI}$ with Py-GC-MS-obtained aromatics and aliphatics do not contradict with the fact that $Se^{VI}$ is the oxidation product of various emitted Se species. Indeed, $Se^{IV}$ and $Se^{VI}$ are expected to undergo redox reactions in the atmosphere as indicated in the relationship between inorganic Se species and pH in precipitation and cloud water (Fig. 6a, P25). We further highlighted this in the conclusions on P28, L687-690:

*"Nevertheless, it should be noted that inorganic Se species ($Se^{IV}$ or $Se^{VI}$) are also affected by atmospheric processing, such as atmospheric oxidative transformations and pH changes, thereby losing their source information with increasing distance to emission sources. Therefore, it is likely that $Se^{IV}$ indicates more local sources, whereas $Se^{VI}$ could also be transported over longer distances."*

**Figure 6a:** could the relationship be due to chemical transformations or/and to partitioning?

The relationship between inorganic Se species and pH is most likely affected by contributing air masses and their associated chemical signatures and conditions, which may include chemical transformations (e.g., redox reactions) and partitioning to aerosol phases. However, our study cannot conclusively identify the contribution of specific processes as we focused on the end products of atmospheric transport, i.e., deposition.

**Technical corrections**

Line 204: "Se" is missing in the list just before Br. done
* * *
**References**

Feinberg, A., Stenke, A., Peter, T., and Winkel, L. H. E.: Constraining Atmospheric Selenium Emissions Using Observations, Global Modeling, and Bayesian Inference, Environmental Science & Technology, 54, 7146-7155, 10.1021/acs.est.0c01408, 2020a.

Feinberg, A., Maliki, M., Stenke, A., Sudret, B., Peter, T., and Winkel, L. H. E.: Mapping the drivers of uncertainty in atmospheric selenium deposition with global sensitivity analysis, Atmos. Chem. Phys., 20, 1363-1390, 10.5194/acp-20-1363-2020, 2020b.

Lao, I. R., Feinberg, A., and Borduas-Dedekind, N.: Regional Sources and Sinks of Atmospheric Particulate Selenium in the United States Based on Seasonality Profiles, Environmental Science & Technology, 57, 7401-7409, 10.1021/acs.est.2c08243, 2023.

Subbalakshmi, Y., Patti, A. F., Lee, G. S. H., and Hooper, M. A.: Structural characterisation of macromolecular organic material in air particulate matter using Py-GC-MS and solid state C-NMR, Journal of Environmental Monitoring, 2, 561-565, 10.1039/B005596O, 2000.

Zhao, J., Peng, P. a., Song, J., Ma, S., Sheng, G., and Fu, J.: Characterization of organic matter in total suspended particles by thermodesorption and pyrolysis-gas chromatography-mass spectrometry, Journal of Environmental Sciences, 21, 1658-1666, https://doi.org/10.1016/S1001-0742(08)62470-5, 2009.

Zhao, J., Peng, P. a., Song, J., Ma, S., Sheng, G., Fu, J., and Yuan, D.: Characterization of Humic Acid-like Substances Extracted from Atmospheric Falling Dust Using Py-GC-MS, Aerosol and Air Quality Research, 12, 83-92, 10.4209/aaqr.2011.06.0086, 2012.

---

## Author Comment (AC3)

**Author response to all referees' comments on "Influences of sources and weather dynamics on atmospheric deposition of Se species and other trace elements"**

We would like to thank the three reviewers for their time reviewing our manuscript and their valuable comments, which significantly enhanced the clarity of the paper. We have considered all comments carefully and present our point-by-point responses below (reviewer comments are in blue and author responses are in black). In the manuscript and supplementary information, the changes are shown using the track changes feature of word processor.

**Response to Referee #2**

I recommend the authors write a shorter abstract. Please, introduce a more summarized writing where you highlight the main findings of the article. I think this paper presents a very complete analysis, but if the authors abbreviate this section, it makes it more appealing to the reader.

Thank you for your feedback. We revised the abstract to make it shorter.

Line 37: The authors state that trace elements and isotopes in water are decoupled from clouds during precipitation. Can the authors indicate the specific result where this effect is observed?

The potential links between trace elements and water isotopes are discussed in details in section 3.1.2, on P17-18, L431-445. For clarification, we adapted the sentence in the abstract on P2, L34-36:

*"Correlations between cloud water isotopes and trace elements indicate that the water and trace element cycles are coupled from the source to the formation of clouds, with possible decoupling during precipitation in relation to below cloud scavenging."*

This is a very long article, as it presents different subsections. I recommend introducing a table of contents at the beginning of the manuscript.

Thank you for your proposition. We asked the editor from ACP and having a table of contents is very uncommon for research articles in ACP. Therefore, after introducing the research questions in the introduction, we indicated now explicitly in what sections they are addressed:

*"The research questions (1), (2) and (3) are discussed in Sect. 3.1, while Sect. 3.2 is focused on the last research question (4).*" (see P6, L131-132)

Methods section: the reviewer strongly recommends the authors extend the description of the study region, specifically to detail the Pic du Midi Observatory monitoring station site. It would be interesting to know the height of the mixing layer in the region. Since this study deals with a synoptic transport of Se, it would be interesting to know if the authors discarded the local influence (convective transport in the mixed layer).

Pic du Midi Observatory is an established long-term monitoring site of different chemical compounds (e.g. ozone, mercury, carbon monoxide, formaldehyde), which included detailed descriptions of the monitoring station (Marenco et al., 1994; Chevalier et al., 2007; Fu et al., 2016; Chevalier et al., 2008; Prados-Roman et al., 2020). Furthermore, extensive research has been conducted to investigate specific influences to the measurements at the monitoring station from the boundary layer (e.g. Gheusi et al. (2011); Hulin et al. (2019)). The typical summer afternoon mixing layer height is 2km (Gheusi et al.,

2011). We did not discard the local influence, because aerosol sampling included these hours. We discuss on P15, L373-374 how higher summertime aerosol Se is likely caused by boundary layer sources and upslope winds.

We agree with the reviewer and included references to previous studies that included a detailed description on the monitoring station on P7, L139-142:

*"This high-altitude monitoring station is only occasionally influenced by the boundary layer through convection or by anabatic (i.e. upslope or valley) wind systems (Gheusi et al., 2011; Hulin et al., 2019), making it particularly suitable to investigate long-range transport as it is mostly exposed to free tropospheric air (Marenco et al., 1994; Henne et al., 2010; Fu et al., 2016)."*

Local influences on Se are described in more detail in the last questions of Referee #2 on Line 365.

From your results: it is possible to assess if Se concentrations decline with precipitation amount?

Se concentrations did not significantly decrease with increasing precipitation amount. The effect of precipitation amount on the concentrations of Se and other elements were explored using principal component analysis (PCA), for which the output is described in detail in the Supplement, S8 (Fig. S11, P20).

For clarification, we added the following sentence on P18, L456-458:

*"While many major and trace elements (e.g. Na, K, P, Fe, Cu, Zn, As, Pb) showed a significant negative correlation with precipitation ($p<0.05$), Se concentrations did not significantly decrease with increasing precipitation amount."*

What about the solubility of Se species: your study, only measures dilute Se species?

Our study focuses on dissolved species in both wet deposition (dissolved fraction filtered with 0.2 µm) and in the water-soluble fraction of aerosols.

The investigation of other Se species, which are not soluble in water, would require specific extraction methods and/or direct solid phase speciation techniques, which have not been developed or are not suitable for low Se concentrations in aerosols collected in remote areas (e.g. synchrotron-based X-ray spectroscopy: detection limits in the $mg \cdot kg^{-1}$ range). From what is known from other matrices (e.g., soils, sediments), it is however likely that $Se^{VI}$ is more water-soluble than $Se^{IV}$, because $Se^{IV}$ is known to be more strongly bound to minerals such as Fe (oxy)hydroxides (Tolu et al., 2014).

In the conclusions, you state that: "for the first time we were able to identify an organic Se species as a biomarker of marine biogenic sources". Could you describe the specific result that allows you to conclude this?

This point was brought up by Referee #1 as well, we replaced the sentence "*for the first time we could identify an organic Se species as a biomarker for marine biogenic sources*" as follows: "*for the first time we could detected a new Se fraction likely of organic nature, which appears to be a biomarker for marine biogenic sources. Further work is required to investigate the molecular composition of this Se fraction and its role in atmospheric Se cycling.*" on P28, L683-685.

This conclusion is based on: i) the detection of an unknown Se peak ("OrgSe) at the same retention time than SeCys$_2$ (described in section 3.2.1, P22, L536-341); and ii) the significant correlations between OrgSe in different atmospheric samples and independent data sets, including the S species DMSO$_2$ and MSA (oxidation production of DMS), the quantitative contribution of Atlantic moisture sources and Chl-*a* exposure, and the proportions of pyrolytic products that are specifically derived from proteins and amino acids. These correlations are discussed in detail in section 3.2.3 (P26-28).

Line 365: Se can be released to the atmosphere from a variety of sources (e.g., natural sources such as soil, water, and vegetation, as well as anthropogenic sources). Are the terrestrial contributions of Se at Pic du Midi only due to synoptic transport or to local sources?

The different applied chemical analyses and modelling techniques include all potential source signatures (including terrestrial contribution) of both long-range and local emission. Specifically, the moisture source analysis included different pre-defined regions, including contribution over e.g., all European countries, North Africa, but also "local" moisture uptakes (in a 0.5° radius around Pic du Midi Observatory). None of the Se species or chemical source signatures correlated significantly to local moisture uptakes (sources), thus suggesting longer-range sources. Although there is no significant correlation between Se species and local moisture uptakes, local sources e.g., due to wet scavenging cannot be fully excluded.

**References**

Chevalier, A., Gheusi, F., Attié, J. L., Delmas, R., Zbinden, R., Athier, G., and Cousin, J. M.: Carbon monoxide observations from ground stations in France and Europe and long trends in the free troposphere, Atmos. Chem. Phys. Discuss., 2008, 3313-3356, 10.5194/acpd-8-3313-2008, 2008.

Chevalier, A., Gheusi, F., Delmas, R., Ordóñez, C., Sarrat, C., Zbinden, R., Thouret, V., Athier, G., and Cousin, J. M.: Influence of altitude on ozone levels and variability in the lower troposphere: a ground-based study for western Europe over the period 2001–2004, Atmos. Chem. Phys., 7, 4311-4326, 10.5194/acp-7-4311-2007, 2007.

Fu, X., Maruszczak, N., Wang, X., Gheusi, F., and Sonke, J. E.: Isotopic Composition of Gaseous Elemental Mercury in the Free Troposphere of the Pic du Midi Observatory, France, Environmental Science & Technology, 50, 5641-5650, 10.1021/acs.est.6b00033, 2016.

Gheusi, F., Ravetta, F., Delbarre, H., Tsamalis, C., Chevalier-Rosso, A., Leroy, C., Augustin, P., Delmas, R., Ancellet, G., Athier, G., Bouchou, P., Campistron, B., Cousin, J. M., Fourmentin, M., and Meyerfeld, Y.: Pic 2005, a field campaign to investigate low-tropospheric ozone variability in the Pyrenees, Atmospheric Research, 101, 640-665, https://doi.org/10.1016/j.atmosres.2011.04.014, 2011.

Henne, S., Brunner, D., Folini, D., Solberg, S., Klausen, J., and Buchmann, B.: Assessment of parameters describing representativeness of air quality in-situ measurement sites, Atmos. Chem. Phys., 10, 3561-3581, 10.5194/acp-10-3561-2010, 2010.

Hulin, M., Gheusi, F., Lothon, M., Pont, V., Lohou, F., Ramonet, M., Delmotte, M., Derrien, S., Athier, G., Meyerfeld, Y., Bezombes, Y., Augustin, P., and Ravetta, F.: Observations of Thermally Driven Circulations in the Pyrenees: Comparison of Detection Methods and Impact on Atmospheric Composition Measured at a Mountaintop, Journal of Applied Meteorology and Climatology, 58, 717-740, 10.1175/jamc-d-17-0268.1, 2019.

Marenco, A., Gouget, H., Nédélec, P., Pagés, J.-P., and Karcher, F.: Evidence of a long-term increase in tropospheric ozone from Pic du Midi data series: Consequences: Positive radiative forcing, Journal of Geophysical Research: Atmospheres, 99, 16617-16632, https://doi.org/10.1029/94JD00021, 1994.

Prados-Roman, C., Fernández, M., Gómez-Martín, L., Cuevas, E., Gil-Ojeda, M., Maruszczak, N., Puentedura, O., Sonke, J. E., and Saiz-Lopez, A.: Atmospheric formaldehyde at El Teide and Pic du Midi remote high-altitude sites, Atmospheric Environment, 234, 117618, https://doi.org/10.1016/j.atmosenv.2020.117618, 2020.

Tolu, J., Thiry, Y., Bueno, M., Jolivet, C., Potin-Gautier, M., and Le Hécho, I.: Distribution and speciation of ambient selenium in contrasted soils, from mineral to organic rich, Science of The Total Environment, 479-480, 93-101, https://doi.org/10.1016/j.scitotenv.2014.01.079, 2014.